# ProTo: Program-Guided Transformer for Program-Guided Tasks

**Zelin Zhao**[*]
The Chinese University of Hong Kong
zelin@link.cuhk.edu.hk

**Karan Samel**
Georgia Institute of Technology
ksamel@gatech.edu

**Binghong Chen**
Georgia Institute of Technology
binghong@gatech.edu

**Le Song**
Biomap and MBZUAI
dasongle@gmail.com

## Abstract

Programs, consisting of semantic and structural information, play an important role in the communication between humans and agents. Towards learning general program executors to unify perception, reasoning, and decision making, we formulate program-guided tasks which require learning to execute a given program on the observed task specification. Furthermore, we propose **Pro**gram-Guided **T**ransf**o**rmer (ProTo), which integrates both semantic and structural guidance of a program by leveraging cross-attention and masked self-attention to pass messages between the specification and routines in the program. ProTo executes a program in a learned latent space and enjoys stronger representation ability than previous neural-symbolic approaches. We demonstrate that ProTo significantly outperforms the previous state-of-the-art methods on GQA visual reasoning and 2D Minecraft policy learning datasets. Additionally, ProTo demonstrates better generalization to unseen, complex, and human-written programs.

## 1 Introduction

Programs are the natural interface for the communication between machines and humans [11]. In comparison to instructing machines via demonstrations [41, 52, 73, 6] or via natural language [14, 32, 3], guiding agents by programs has multiple benefits. First, programs are explicit and much cleaner than other instructions such as languages [81]. Second, programs are structured with loops and branches [1] so they can express complex reasoning processes [99]. Finally, programs are compositional, promoting the generalization and scalability of neural models [15, 64, 20]. However, while program synthesis and program induction have been deeply explored [12, 17, 18, 22, 39, 49], very few works focuses on learning to follow program guidance [81, 71]. Furthermore, previous work designs ad-hoc program executors for different functions in different tasks [81, 29, 23], which hinders the generalization and scalability of developed models.

To pursue general program executors to unify perception, reasoning, and decision making, we formulate *program-guided tasks*, which require the agent to follow the given program to perform tasks conditioned on task specifications. The programs may come from a program synthesis model [99] or be written by human [81]. Two exemplar tasks are shown in Figure 1. Program-guided tasks are challenging because the agent needs to jointly follow the complex structure of the program [1], perceive the specification, and ground the program semantics on the specification [40].

Inspired by the recent significant advance of transformers in diverse domains [24, 25, 59], we present the Program-guided Transformer (ProTo) for general program-guided tasks. ProTo combines the

35th Conference on Neural Information Processing Systems (NeurIPS 2021).

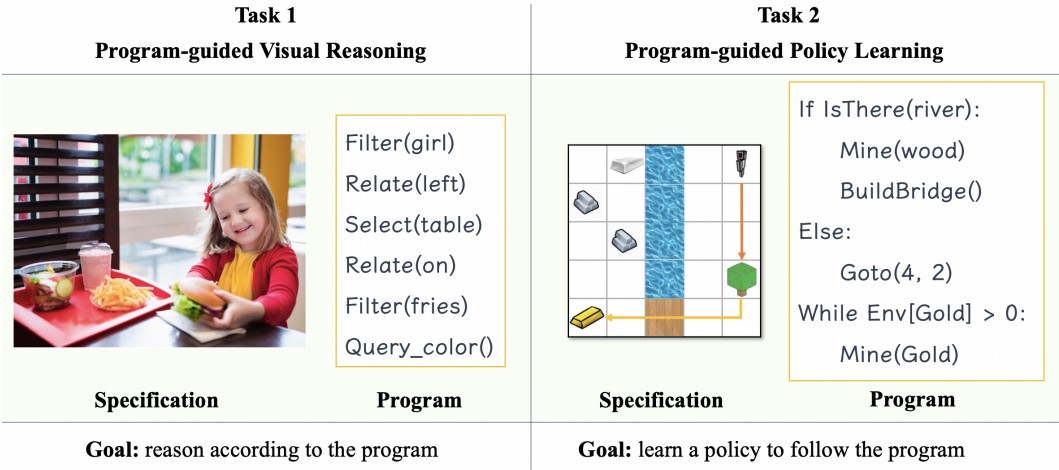

| Task 1 | Task 2 |
| Program-guided Visual Reasoning | Program-guided Policy Learning |

| Specification | Program | Specification | Program |

**Goal:** reason according to the program | **Goal:** learn a policy to follow the program

Figure 1: Illustration of two exemplar program-guided tasks. The first task is program-guided visual reasoning [47], where the model needs to learn to execute a visual program on the specification (image) to get a predicted answer. The second task is program-guided policy learning [81], where the agent learns a policy to perform tasks based on the observed specification following the program guidance.

strong representation ability of transformers with symbolic program control flow. We propose to separately leverage the program semantics and the explicit structure of the given program via efficient attention mechanisms. ProTo enjoys strong representation ability by executing the program in a learned latent space. In addition, ProTo can either learn from reward signals [64, 81] or from dense execution supervision [15].

We evaluate ProTo on two tasks, program-guided visual reasoning and program-guided policy learning (corresponding to Figure 1 left and Figure 1 right). The former task requires the model to learn to reason over a given image following program guidance. We experiment on the GQA dataset [47] where the programs are translated from natural language questions via a trained program synthesis model. The evaluation on the public test server shows that we outperform the previous state-of-the-art program-based method by 4.31% in terms of accuracy. Our generalization experiments show that ProTo is capable of following human program guidance. The latter task asks the agent to learn a multi-task policy to interact with the environment according to the program. We experiment on a 2D Minecraft environment [81] where the programs are randomly sampled from the domain-specific language (DSL). We find that ProTo has a stronger ability to scale to long and complex programs than previous methods. ProTo significantly outperforms the vanilla transformer [87] which does not explicitly leverage the structural guidance of the program. We will release the code and pre-trained models after publishing.

In summary, our contributions are threefold. First, we formulate and highlight program-guided tasks, which are generalized from program-guided visual reasoning [99, 64] and program-guided policy learning [81]. Second, we propose the program-guided transformer for program-guided tasks. Third, we conduct extensive experiments on two domains and show promising results towards solving program-guided tasks.

## 2   Related Work

**Program Induction, Synthesis and Interpretation**    Program induction methods learn an underlying input-output mapping from given examples [54, 22, 50], and no programs are explicitly predicted. Differently, program synthesis targets at predicting symbolic programs from task specifications [23, 12, 18, 17, 31, 27, 93, 85, 16, 80, 58, 86]. However, these approaches do not learn to execute programs. In the domain of digit and string operation (adding, copying, sorting, etc.), Neural Program Interpreter (NPI) [71, 94, 75] learn to compose lower-level programs to express higher-level programs [37] while Neural Turing Machines [38, 55] use attentional processes with external memory to infer simple algorithms. Recently, [81] proposes to guide the agent in 2D Minecraft via programs. [65]

detects repeated entries in the image and uses programs to guide image manipulation. They neither attempt to formulate general program-guided tasks nor propose a unified model for different tasks.

**Visual Reasoning**    Visual reasoning requires joint modeling of natural language and vision. A typical visual reasoning task is the visual question answering (VQA) [5]. Attention mechanisms have been widely used in the state-of-the-art VQA models [60, 2, 74, 104, 46, 72]. Neural module networks and the neural-symbolic approaches [99, 64] propose to infer programs from natural languages and execute programs on visual contents to get the answer. A large-scale dataset GQA [47], which addresses the low diversity problem in the synthetic CLEVR dataset [48], is proposed to evaluate the state-of-the-art visual reasoning models. Neural state machine [45] is proposed by leveraging modular reasoning over a probabilistic graph, which does not leverage the symbolic program. Our model is similar to the concurrent work, meta module network [15] in that we use shared parameters for different function executors. Nevertheless, we propose a novel attention-based architecture and extend our model to the policy learning domain.

**Transformer**    Transformer was firstly proposed in machine translation [87]. After that, transformers demonstrate their power on many different domains, such as cross-modal reasoning [83], large-scale unsupervised pretraining [24, 79, 98], and multi-task representation learning [70, 67]. Recently, transformers appear to be competitive models in many fundamental vision tasks, such as image classification [25, 101, 66], object detection [13] and segmentation [59, 33]. While we share the same belief that transformers are strong, unified, elegant models for various deep learning tasks, we propose novel transformer architecture for program-guided tasks. We discuss a few similar unpublished works in Appendix A.

**Policy Learning With Programs**    Previous policy learning literature explored the benefits of programs in different aspects. First, some pre-defined routines containing human prior knowledge could help reinforcement learning, and planning [30, 68, 4, 103]. Second, programs enable interpretable and verifiable reinforcement learning [89, 8]. Our approach follows a recent line of work that learns to interpret program guidance [81, 4]. However, we adopt a novel, simple and uniform architecture that demonstrates superior performance, thus conceptually related to multitask reinforcement learning [84, 92] and hierarchical reinforcement learning [7].

## 3    Program-guided Tasks

Unlike natural language that is flexible, complex, and noisy, programs are structured, clean and formal [81]. Therefore, programs can serve as a powerful intermediate representation for human-computer interactions [11]. Different from previous work that *learns to synthesis* programs from data [99, 85, 27], we study *learning to execute* given programs [81] based on three motivations. First, instructing machines via explicit programs instead of noisy natural languages enjoys better efficiency, and accuracy [81]. Second, learned program executors have stronger representation ability than hard-coded ones [99, 15]. Third, we attempt to develop a unified model to integrate perception, reasoning, and decision by learning to execute, which heads towards a core direction in the neural-symbolic AI [36, 35].

A program-guided task is specified by a tuple $(\mathcal{S}, \Gamma, \mathcal{O}, \mathcal{G})$ where $\mathcal{S}$ is the space of specifications, $\Gamma$ denotes the program space formed by a domain-specific language (DSL) for a task, $\mathcal{O}$ is the space for execution results, and $\mathcal{G}$ is the goal function for the task. For each instance of the program-guided task, a program $P \in \Gamma$ is given. An executor $\mathcal{E}_\Phi$ is required by the task to execute the program on the observed specification. Note that different from hard-coded non-parametric executors used in some previous work [99, 23], the executor $\mathcal{E}_\Phi$ is parametrized by $\Phi$. For convenience, we define a routine to be the minimum execution unit of a program (e.g. `Filter(girl)` in Figure 1 is a routine). We denote $P_k$ to be the $k$-th routine in $P$ and $|P|$ to be the number of routines in $P$. According to the DSL, the program should begin at the *entrance routine* and finish at one of the *exiting routines* (e.g. `Filter(girl)` in Figure 1 is an entrance routine and `Query_color()` is an exiting routine). At each execution step $\tau$, the executor executes the current routine $P_p \in P$ on the specification $s^{(\tau)} \in \mathcal{S}$ and produces an output $o^{(\tau)} \in \mathcal{O}$. The execution of $P$ finishes if one of the exiting routines finishes execution. The goal of a program-guided task is to produce desired execution results to achieve the goal $\mathcal{G}$.

| **Algorithm 1:** ProTo Execution | **Algorithm 2:** UpdatePointer($p$, $o^{(\tau)}$, $P_p$) |
|---|---|
| **Result:** Execution results $\{o^{(\tau)}\}_{\tau=1}^{T}$
1 Initialize $\tau = 0$ and the pointer $p$;
2 Build $\mathbf{P}^s$ according to Eq. 1;
3 **while** *not reach an exiting routine* **do**
4     Observe $s^{(\tau)}$ and set $\tau = \tau + 1$ ;
5     Build $\mathbf{P}^{t(\tau)}$ according to Eq. 2;
6     $o^{(\tau)} = \text{ProToInfer}(s^{(\tau)}, \mathbf{P}^s, \mathbf{P}^{t(\tau)}, p)$;
7     Output $o^{(\tau)}$ ;
8     **if** $P_p$ *finishes execution* **then**
9        UpdatePointer($p$, $o^{(\tau)}$, $P_p$) ;
10 **end** | 1 **if** $P_p$ *is an* `If`*-routine* **then**
2     Point $p$ to the first routine in the $T/F$
      branch if $o^{(\tau)}$ is $T/F$;
3 **else if** $P_p$ *is a* `While`*-routine* **then**
4     Point $p$ to the first routine
      inside/outside the loop if $o^{(\tau)}$ is $T/F$;
5 **else if** $P_p$ *ends a loop* **then**
6     Point $p$ to the loop condition routine.
7 **else**
8     Point $p$ to the subsequent routine;
9 **end** |

## 4 Program-guided Transformer

A good model for a program-guided task should fulfill the following merits. First of all, it should leverage both the semantics and structures provided by the program (refer to Figure 2). Second, it would better be a multi-task architecture with shared parameters for different routines to ensure efficiency and generalization [15, 90, 70]. Third, it would be better if the model does not leverage external memory for the efficiency of batched training [64]. To these ends, we present the Program-guided Transformer (ProTo). ProTo treats *routines as objects* and leverages attention mechanisms to model interactions among routines and specifications. The execution process of ProTo is presented in Algorithm 1. In each execution timestep, ProTo leverages the semantic guidance $\mathbf{P}^s$ and the current structure guidance $\mathbf{P}^{t(\tau)}$ (Line 2&5 in Algorithm 1, detailed in Sec 4.1) and infers for one step (Line 6 in Algorithm 1, detailed in Sec 4.2). When a routine finishes execution, a pointer $p \in \{1, 2, ..., |P|\}$ indicating the current routine $P_p$ ($P_p$ is the $p$th routine in $P$) would be updated (Line 8&9 in Algorithm 1, detailed in Algorithm 2 and Sec 4.3). The results of ProTo are denoted as $\{o^{(\tau)}\}_{\tau=1}^{T}$ where $T$ is the total number of execution timesteps. ProTo supports several training schemes, which are listed in Sec 4.4.

### 4.1 Disentangled Program Representation

As shown in the left of Figure 2, we leverage both the semantics and structure of the program in ProTo. The semantic part of the program $\mathbf{P}^s \in \mathbb{R}^{|P| \times d}$ is an embedding matrix for all routines where $\mathbb{R}$ is the real coordinate space and $d$ is a hyperparameter specifying the feature dimension. Note $\mathbf{P}^s$ remains the same for all execution timesteps. Denote $\{\mathbf{w}_i^k\}_{i=0}^{L}$ to be the words in $P_k$ where $L$ is the maximum number of words in one routine (padding words are added if $P_k$ does not have $L$ words), we construct the $k$-th row of $\mathbf{P}^s$ corresponding to $P_k$ via a concatenation of all token embeddings:

$$\mathbf{P}_k^s = \big\Vert_i \text{WordEmbed}(\mathbf{w}_i^k) \tag{1}$$

where $\Vert \cdot$ represents the concatenation function and WordEmbed maps a token to an embedding with dimension $d_m$ where we set $d = L \times d_m$.

The structure part of the program $\mathbf{P}^{t(\tau)} \in \mathbb{R}^{|P| \times |P|}$ is a transition mask calculated at each execution timestep $\tau$ to pass messages from the previous execution timestep to the current execution timestep. Although most types of the routines only need to get information from the previously executed routines, some logical routines such as `Compare_Color()` rely on more than one routines' results. We denote $\text{Parents}(P_p)$ to be the set of routines whose results would be taken as input by the current routine $P_p$. We can easily derive $\text{Parents}(P_p)$ from the program $P$, which is detailed in the Appendix C.1. The transition mask $\mathbf{P}^{t(\tau)}$ is defined by the following equation:

$$\mathbf{P}^{t(\tau)}[i][j] = \begin{cases} 0 & \text{if } (P_j \in \text{Parents}(P_p) \text{ and } P_i = P_p) \text{ or } (i = j), \\ -\infty & \text{else,} \end{cases} \tag{2}$$

where we set diagonal elements of $\mathbf{P}^{t(\tau)}$ to zeros to preserve each routine's self information to the next timestep. Positional encoding is added following the standard transformer [87].

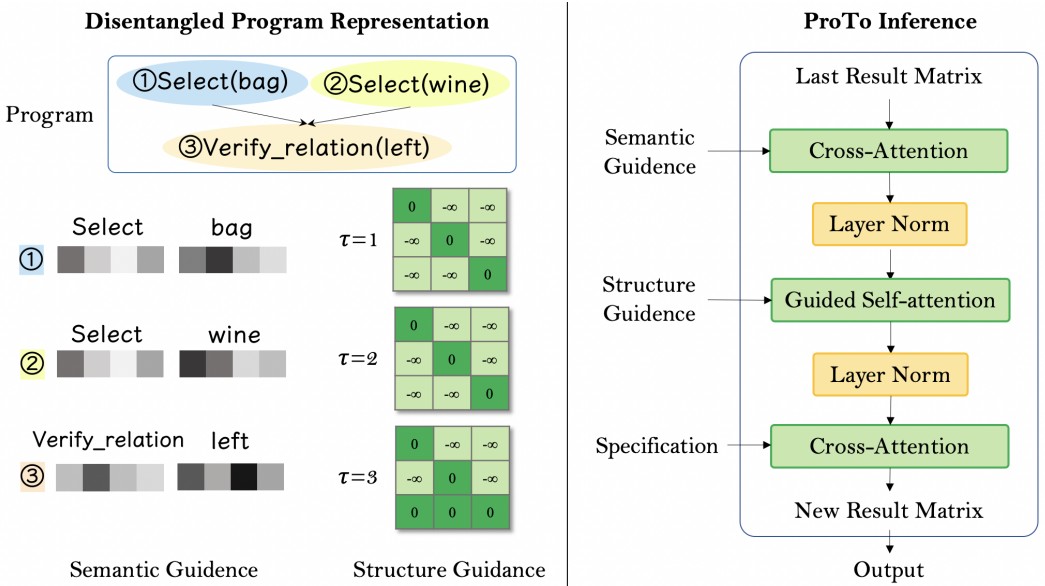

Figure 2: Main components of Program-guided Transformer (ProTo). Left: we propose a disentangled program representation to leverage both semantics and structures of the program. The shown program selects the bag and the wine from the image and verifies whether the bag is on the left to the wine. Right: given structure and semantic guidance offered by a program, ProTo updates the result matrix leveraging the program guidance and the specification.

## 4.2 ProTo Inference

Given the disentangled program guidance $\mathbf{P}^s$, $\mathbf{P}^{t(\tau)}$ and the current observed specification $s^{(\tau)} \in \mathbb{R}^{N_S \times d}$ where $N_s$ is the number of objects in the specification, ProTo infers the execution result $o^{(\tau)}$ via stacked attention blocks as shown on the right of Figure 2, which is presented in detail in this subsection. Note we assume the specification is represented in an object-centric manner, which might be obtained via a pre-trained object detector [2] or an image patch encoder [25].

ProTo maintains a result embedding matrix $\mathbf{Z} \in \mathbb{R}^{|P| \times d}$ which stores latent execution results for all routines. The result embedding matrix is initialized as a zero matrix and is updated at each timestep. We show the architecture of transformer executors in the right of Figure 2. During ProTo inference, we first conduct cross-attention to inject semantic information of routines to the hidden results via

$$\mathbf{H_1} = \text{CrossAtt}_1(\mathbf{Z}, \mathbf{P}^s) = \text{softmax}(\frac{\mathbf{Z}\mathbf{P}^{s\top}}{\sqrt{d}})\mathbf{P}^s \tag{3}$$

where $\mathbf{H_1} \in \mathbb{R}^{|P| \times d}$ is the first intermediate latent results.

We propose to adopt masked self-attention [87] to propagate the information of the previous results to the current routine. The masked self-attention leverages the transition mask to restrict the product of queries and keys before softmax (queries, keys and values are the same in the self-attention). Formally, we acquire the second intermediate results $\mathbf{H_2} \in \mathbb{R}^{|P| \times d}$ by

$$\mathbf{H_2} = \text{MaskedSelfAtt}(\mathbf{H_1}, \mathbf{P}^{t(\tau)}) = \text{softmax}(\frac{\mathbf{H_1}\mathbf{H_1}^\top}{\sqrt{d}} + \mathbf{P}^{t(\tau)})\mathbf{H_1}. \tag{4}$$

After that, we apply cross-attention again to update results based on the specification $s^{(\tau)}$:

$$\mathbf{Z} = \text{CrossAtt}_2(\mathbf{H_2}, s^{(\tau)}) = \text{softmax}(\frac{\mathbf{H_2}s^{(\tau)\top}}{\sqrt{d}})s^{(\tau)}. \tag{5}$$

The above equations for self-attention and cross-attention show only one head of self-attention and cross-attention for simplicity. However, in experiments, we use multi-head attention with eight heads [87]. The resulting embedding corresponding to the pointed routine $\mathbf{Z}_p$ would be decoded via an MLP to produce an explicit output $o^{(\tau)} = \text{MLP}(\mathbf{Z}_p)$.

### 4.3 Pointer Update

After each execution timestep, we would update the pointer $p$ if $P_p$ finishes execution. Note that some routines cannot be finished in one execution timestep (e.g., the routine `Mine(gold)` requires the agent to navigate to the gold and then take `Mine` action). We know the execution of $P_p$ is finished if the execution result $o^{(\tau)}$ meets the ending condition of $P_p$ (e.g. when $o^{(\tau)}$ is the `Mine` action and the agent successfully mines a gold, we judge that `Mine(gold)` is finished). A full list of ending conditions for all routines is in the Appendix B.

The pointer $p$ is updated according to the control flow [1, 18, 81] of the program. The pointer's movement is determined by the execution result $o^{(\tau)}$, the type of $P_p$, and the location of $P_p$. We outline the detailed procedure of pointer update in Algorithm 2.

**Parallel Execution** Transformer has one powerful ability that it can conduct sequential prediction in parallel [87]. In our case, we can execute routines that do not have result dependency in parallel. We only need to modify the masked self-attention in Eq. 4 to support passing multiple routines' messages in parallel. Furthermore, multiple pointers would be adopted and updated in parallel, while multiple results would be output in parallel in one execution timestep. We detail the paralleled version of Algorithm 1 and Algorithm 2 in the Appendix C.2.

### 4.4 ProTo Training Targets

After deriving the execution results of all routines denoted via $\{o^{(\tau)}\}_{\tau=1}^{T}$, ProTo can be trained via the following three types of training targets. (1) Dense Supervision $\mathcal{L}_D$. When the ground truth of all execution results of all routines $\{\widetilde{o}^{(\tau)}\}_{\tau=1}^{T}$ is known, we can use a dense loss $\mathcal{L}_D$, which is the $L_2$ distance between $\{o^{(\tau)}\}_{\tau=1}^{T}$ and $\{\widetilde{o}^{(\tau)}\}_{\tau=1}^{T}$. (2) Partial Supervision $\mathcal{L}_P$. Knowing the final execution result of the whole program $\widetilde{o}^{(T)}$, ProTo can be learned through partial supervision $\mathcal{L}_P$ which measures the $L_2$ distance between $o_T$ and $\widetilde{o}^{(T)}$. (3) RL Target $\mathcal{L}_R$. When a program is successfully executed, the environment gives the agent a sparse reward of $+1$. Otherwise, the agent gets a zero reward. In this case, ProTo can be optimized via a reinforcement learning loss $\mathcal{L}_R$ [51, 81]. In experiments, we follow the same type of supervision as the corresponding baselines, which varies from task to task.

## 5 Experiments

### 5.1 Program-guided Visual Reasoning

**Task Description** Program-guided visual reasoning requires the agent to follow program guidance to reason about an image. It is formed by a tuple $(\mathcal{S}^v, \Gamma^v, \mathcal{O}^v, \mathcal{G}^v)$ where $v$ denotes the visual reasoning task. One specification $s \in \mathcal{S}^v$ is the object-centric representation of an image formed via a pre-trained object detector [2], which is not updated from time to time. In other words, $s^{(\tau)} = s^{(\tau+1)}$ holds for each timestep $\tau$. The design of the program space $\Gamma^v$ follows the previous work [47]. The output set $\mathcal{O}^v$ is the possible results for all routines for all the programs (all types of results are encoded to fixed-length vectors as explained in Appendix D.1). We define an *answer* to a program $P$ to be the final execution result of the program $P$, and the goal of program-guided visual reasoning $\mathcal{G}^v$ is to predict the correct answer to the program. As for the training target, we adopt the dense supervision $\mathcal{L}_D$ as described in Sec 4.4 to train ProTo following previous approaches [15, 42, 56].

**Dataset Setup** We conduct experiments of program-guided visual reasoning based on the public GQA dataset [47] consisting of 22 million questions over 140 thousand images. It is divided into training, validation, and testing splits. The ground true answers, programs, and scene graphs are provided in the training and validation split but not in the test split. We use the provided balanced training split to control data bias. On the training split, we train a transformer-based seq2seq model [87] to parse a question into a program. For validation and testing, we use this trained seq2seq model to acquire a program from a question [2]. Besides answer accuracy, the GQA dataset [47] offers three more metrics evaluating the consistency, validity, and plausibility of learned models.

---

[2]In the GQA dataset [47], we found that a simple seq2seq model can achieve 98.1% validation accuracy to convert a natural language question into a program. The concurrent work [15] also found a similar fact.

Table 1: Comparison of ProTo with previous models on the `test2019` split of the GQA dataset. In the collum of training Signal, we use QA, SG, Prog to denote question-answer pairs, scene graphs, and programs. As for the metrics, Cons., Plaus., Valid., Distr., Acc. represent consistency, plausibility, validity, distribution, and accuracy correspondingly.

| Model | Signal | Binary | Open | Cons. | Plaus. | Valid. | Distr. | Acc. |
|---|---|---|---|---|---|---|---|---|
| Human [47] | - | 91.20 | 87.40 | 98.40 | 97.20 | 98.90 | - | 89.30 |
| BottomUp [2] | QA | 66.64 | 34.83 | 78.71 | 84.57 | 96.18 | 5.98 | 49.74 |
| MAC [46] | QA | 71.23 | 38.91 | 81.59 | 84.48 | 96.16 | 5.34 | 54.06 |
| LXMERT [83] | QA | 77.16 | 45.47 | 89.59 | 84.53 | 96.35 | 5.69 | 60.33 |
| NSM [45] | SG | 78.94 | 49.25 | 93.25 | 84.28 | 96.41 | 3.71 | 63.17 |
| PVR [56] | Prog | 78.02 | 43.75 | 91.43 | 84.77 | 96.50 | 6.00 | 59.81 |
| SNMN [42] | Prog | 73.40 | 40.82 | 85.11 | 84.79 | 96.37 | 5.14 | 56.09 |
| MMN [15] | Prog | 78.90 | 44.89 | 92.49 | 84.55 | 96.19 | 5.54 | 60.83 |
| ProTo | Prog | **79.12** | **51.45** | **93.45** | **86.12** | **96.52** | **3.66** | **65.14** |

Table 2: Accuracy of the ablation study on the validation split of the GQA dataset.

| Model | Acc. |
|---|---|
| ProTo Full Model | **64.47** |
| No Structure Guidance | 55.16 |
| No Semantic Guidance | 33.82 |
| GAT Encoding | 58.58 |
| Partial Supervision | 60.28 |
| NS-VQA [97] | 29.57 |
| IPA-GNN [10] | 47.10 |
| MMN [15] | 60.40 |

Table 3: Results of the generalization experiments on the validation split of GQA dataset.

| Generalization | Model | Acc. |
|---|---|---|
| Human Program Guidance | MMN [15] | 59.47 |
| | ProTo | **70.33** |
| Unseen Programs | MMN [15] | 61.88 |
| | ProTo | **65.34** |
| Restricted Data Regime | MMN [15] | 52.39 |
| | ProTo | **58.31** |

**Experiment Details** We take $N = 50$ object features (provided by the GQA dataset) with $d = 2048$ dimension. The optimizer is BERT Adam optimizer [24] with a base learning rate $1 \times 10^{-4}$, which is decayed by a factor of $0.5$ every epoch. To alleviate over-fitting, we adopt an L2 weight decay of $0.01$. The model is trained for 20 epochs on the training split, and the best model evaluated on the validation split is submitted to the public evaluation server to get testing results. The testing results of baselines are taken from the corresponding published literature [83, 45, 15] or the leaderboard. Following [15, 45], we do not list unpublished methods or methods leveraging external datasets.

**Testing Results** We present the results on the testing split of GQA comparing to previous baselines in Table 1. ProTo surpasses all previous baselines by a considerable margin, successfully demonstrating the effectiveness of leverage program guidance in reasoning. The superior performance of ours over the concurrent work meta module network (MMN) [15] reveals ProTo's strong modeling ability. More visualizations are in Appendix D.1.

**Ablation Study** We ablate our full model to the following variants to study the importance of different components. Despite the described changes, other parts remain the same with the full model. (1) No Structure Guidance. Only the semantic guidance is used, and $\mathbf{P}^{t(\tau)}$ is set to the all-zeros matrix in Eq 4. (2) No Semantic Guidance. The semantic guidance $\mathbf{P}^s$ is set to the all-zeros matrix to disable semantics guidance. (3) GAT Encoding. We use graph attention networks [88] to encode the program and fuse the program feature with result embedding matrix and specification feature via two consecutive cross attention modules (refer to Appendix D.1 for details). (4) Partial Supervision. We only supervise the predicted answer to the program but do not give dense intermediate supervision (refer to Sec 4.4).

Table 2 shows the results, and the validation accuracy of two program-based baselines is listed for reference. The results demonstrate that both structure guidance and semantic guidance contribute significantly to the overall performance. ProTo is also better than the GNN baseline because of the strong cross-modal representation learning ability of the transformer [43]. And the extremely

low validation accuracy of NS-VQA [99], which is reported by its authors in [97], reveals that the hard-coded program executors are not as powerful as the learned transformer executors.

**Generalization Experiments**    We conduct systematical experiments to evaluate whether humans can guide the reasoning process via programs. More details are in the Appendix D.1. (1) Human Program Guidance. We test whether humans can guide the reasoning process via programs on a collected GQA-Human-Program dataset. We ask volunteers to write 500 programs and corresponding answers on 500 random pictures taken from the GQA validation split. No natural language questions are collected. All the models are trained on the training split of GQA and tested on the GQA-Human-Program dataset. (2) Unseen Programs. Following [15], we remove from the training split all the programs containing the function `verify_shape`, and we evaluate the models on the instances containing `verify_shape` on the validation split. (3) Restricted Data Regime. We restrict the models only to use $10\%$ uniformly sampled training data to test the data efficiency of models.

Results are presented on Table 3. We found that ProTo can successfully generalize its learned program execution ability to human written programs, surpassing the previous state-of-the-art neural module network approach by over 10 points. ProTo can also generalize to unseen programs, again verifying the compositional generalization ability of our neural-symbolic transformer model [24, 20]. Besides, ProTo is more data-efficient than MMN [15]. We also found that ProTo is much more effective than the recent learning-to-execute approach IPA-GNN [10].

## 5.2 Program-guided Policy Learning

**Task Description**    Program-guided policy learning requires the agent to learn a policy perform tasks following a given program [81]. We denote this task as a tuple $(\mathcal{S}^p, \Gamma^p, \mathcal{O}^p, \mathcal{G}^p)$ where $p$ stands for the policy learning task. Since we are experimenting on a grid-world environment, the specification $s \in \mathcal{S}^p$ is the feature embeddings of $N_g$ objects placed in $N_g$ grids. Unlike the visual reasoning task, the specification is updated by the environment at each timestep $\tau$ after the agent takes an action. The design of the program space $\Gamma^p$ follows [81]. The output space $\mathcal{O}^p$ consisting of several types: (1) Boolean results (`True` or `False`); (2) Motor Actions (e.g. `Up`); (3) Interactive Actions (e.g. `Mine` and `Build`). Note the agent can only interact with the grid it stands on. In this task, the agent should learn from a reward signal $\mathcal{L}_R$ (described in Sec 4.4) while we also experiment applying the dense supervision. The goal $\mathcal{G}^p$ is to maximize the task completion rates [81].

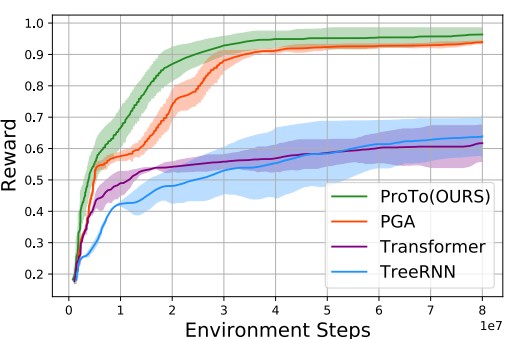

Figure 3: Training curves on the 2D Minecraft environments in comparison to several baselines. Both mean and standard errors over ten random agents are shown in the figure.

**Experiment Details**    Following [77, 81], we conduct experiments on a 2D Minecraft Environment[3]. Programs are sampled from the DSL and divided into training and testing splits (4000 for training and 500 for testing). For each instance, the agent must follow a given program to navigate the grid world, mine resources, sell mined resources from its inventory, or place marks. The baselines include Program-guided Agent (PGA) [81], the naïve Transformer [87], and TreeRNN [82]. PGA separately learns perception and policy modules. Transformer and TreeRNN encode the input program in a token by token manner and output an action distribution. We ensure that the number of parameters for different methods is comparable. We use the same manner of encoding the objects in the grid into features as [81], which are projected to $d$-dimension features via an MLP. The policy is optimized via the actor-critic (A2C) algorithm [51], and we use the same policy learning hyperparameters with PGA [81], which are detailed in the Appendix D.2 for reference. When using dense supervision, the ground-true execution traces come from a hard-coded planner. More details are in the Appendix D.2.

**Training Curves, Testing Results and Visualization**    We first show the training curves under the RL target in Figure 3. We observe that ProTo surpasses Program-guided Agent (PGA) [81],

---

[3]The implementation of the environment, the dataset, and the baselines are provided by the authors of [81].

Table 4: Task completion rates on 2D MineCraft under different test settings. We repeat each experiments on 10 different random seeds and we show both averaged rates and their standard deviations. Baseline numbers are slightly different from [81] due to random noises.

| Supervision | RL Target | | | Dense Supervision | |
|---|---|---|---|---|---|
| Model | Transformer [87] | PGA [81] | ProTo | PGA[81] | ProTo |
| Standard Testing | 50.1±3.2 | 94.2±0.8 | **97.3±2.1** | 96.9±0.9 | **99.1±0.5** |
| Longer Programs | 41.2±3.5 | 86.1±0.9 | **91.2±3.3** | 92.1±0.7 | **94.4±0.8** |
| Complex Programs | 40.8±1.8 | 89.7±0.3 | **95.0±2.5** | 91.2±0.5 | **96.3±1.0** |

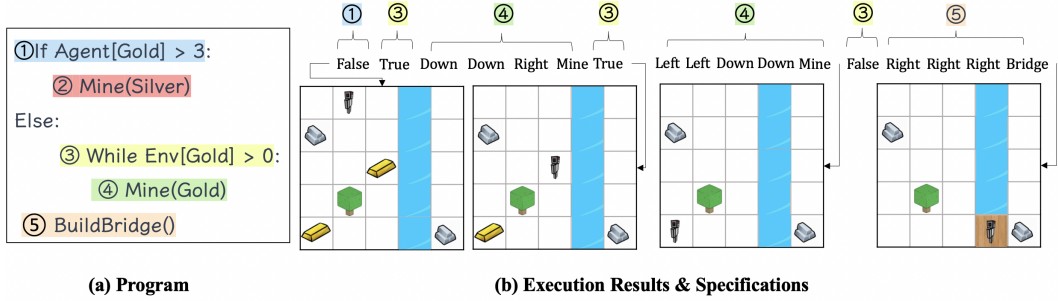

(a) Program

(b) Execution Results & Specifications

Figure 4: An example of result on the 2D Minecraft environment. The given program is shown in (a). Each timestep's execution results and specifications are shown in (b) top and (b) bottom, respectively. Due to space limitations, we only show some specifications along the timeline, and the arrows in (b) denote the places of the specifications in the timeline.

which demonstrates the power of leveraging disentangled program guidance in transformers. The fact that ProTo outperforms the vanilla end-to-end transformer by a large margin demonstrates the effectiveness of explicitly leveraging program structure guidance.

We test the trained agent in different settings. Despite the Standard Testing split offered by [81], we sample two more splits from the DSL while ensuring the testing cases are not seen in the training split: Longer Programs and Complex Programs. All programs in Longer Programs contain more than eighty tokens, while all programs in Complex Programs include more than four `If` or `While` tokens. Furthermore, we add execution losses on the training split and test the baseline method PGA and our method. The results on the test splits are shown in Table 4. We find that ProTo performs better than PGA [81] on all testing settings. ProTo scales better than PGA to longer and complex programs, demonstrating the strong ability of ProTo to leverage the program structure. We also observe that ProTo has superior performance when dense execution supervision is provided. A demonstration of the test split is provided in Figure 4, where we observe ProTo successfully learns to execute the program and develops a good policy to follow the program guidance.

## 6  Conclusion and Future Work

In this paper, we formulated program-guided tasks, which asked the agent to learn to interpret and execute the given program on observed specifications. We presented the Program-guided Transformer (ProTo), which addressed program-guided tasks by executing the program in a hidden space. ProTo provides new state-of-the-art performance versus previous dataset-specific methods in program-guided visual reasoning and program-guided policy learning.

Our work suggests multiple research directions. First, it's intriguing to explore more advanced and challenging program-guided tasks, such as program-guided embodied reasoning [21] and program-guided robotic applications [69, 96, 95, 102]. Second, we are learning separate parameters for different tasks, while building general and powerful program executors across tasks with shared parameters is very promising [44, 92, 90]. Additionally, improving transformer executors with more hierarchy [59] and better efficiency [100] is a meaningful future direction.

# 7 Acknowledgement

We thank Shao-Hua Sun for sharing his codes. This work is supported in part by the DARPA LwLL Program (contract FA8750-19-2-0201).

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
