

Figure 1: Visualization of validation results on the GQA dataset (Part 1). We use the green tick and the red cross to denote right and wrong reasoning result respectively. The MMN approach fails both examples while ProTo successfully addresses them.

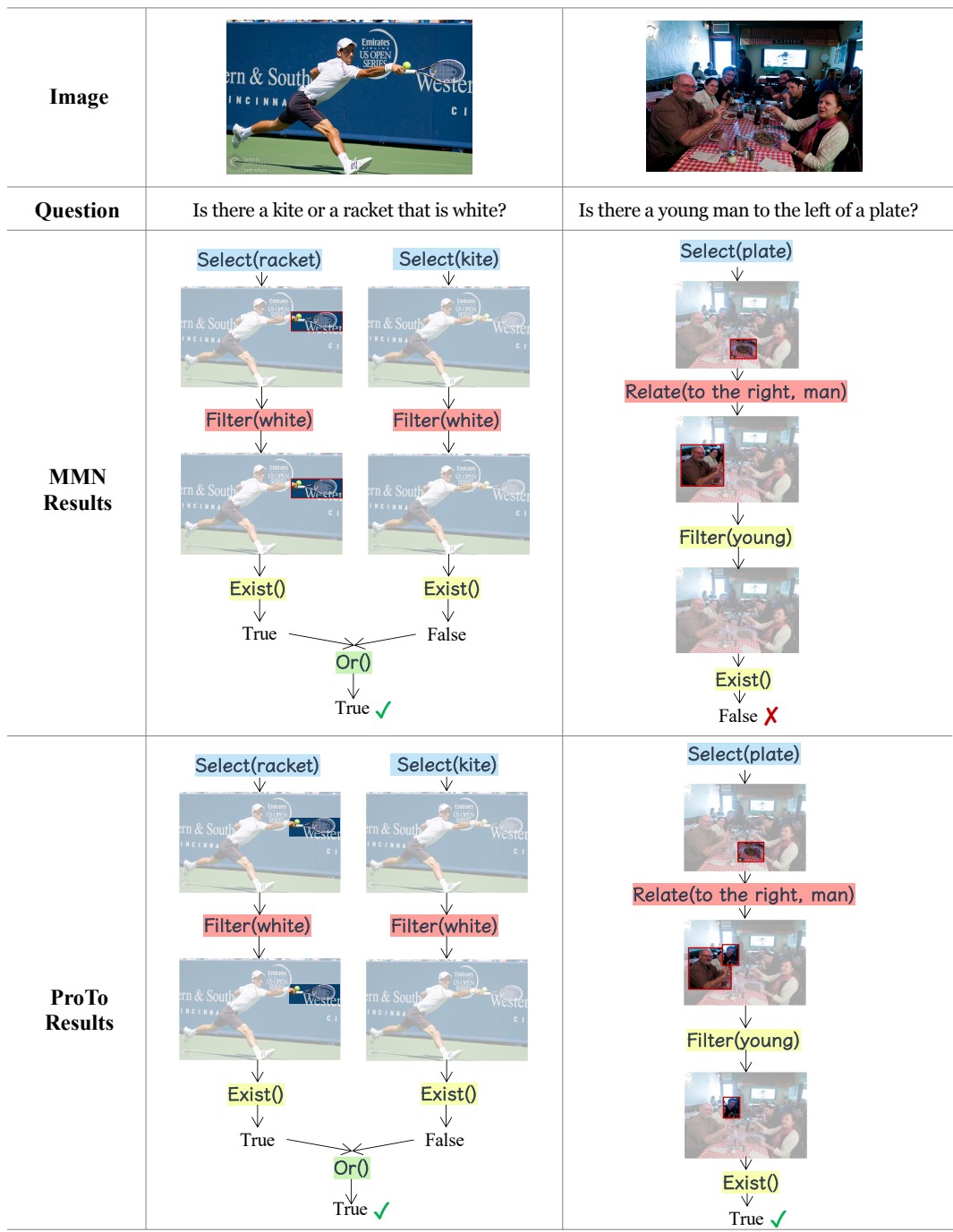

Figure 2: Visualization of validation results on the GQA dataset (Part 2). We use the green tick and the red cross to denote right and wrong reasoning results, respectively. Both models predict the correct answer in the first example, and the reasoning traces are the same. For the second example, our model gets the correct answer, but the MMN baseline fails.

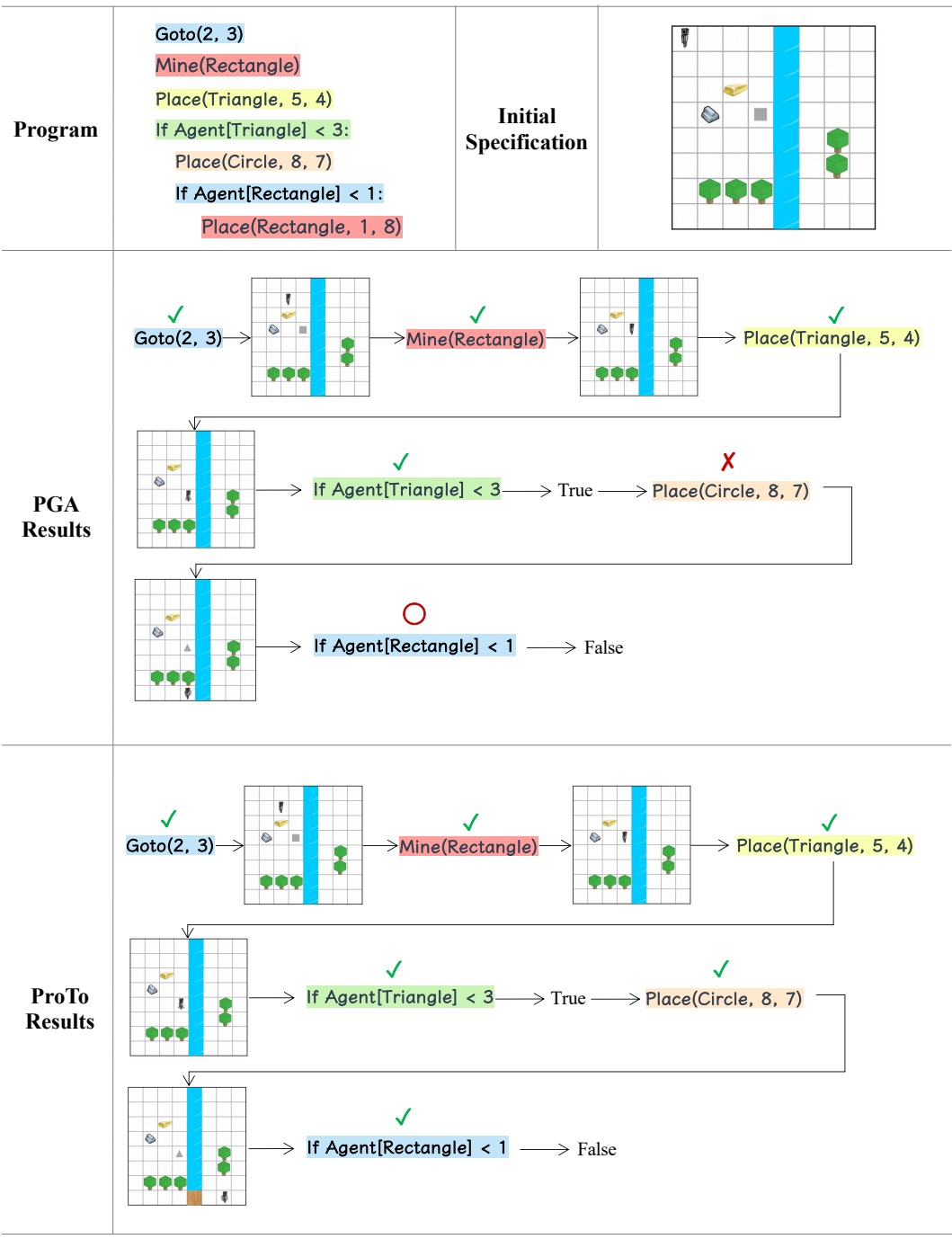

Figure 3: Visualization of test results on the Minecraft environment in comparison to program-guided agent [81] (Part 1). Due to space limitations, we omit the middle steps for executing routines and directly show the execution results of routines. We use the green tick and the red cross to denote right and wrong reasoning results, respectively. Additionally, we use a red circle to indicate that the agent has failed some routine (therefore, the program execution is failed). We observe that the ProTo agent can build a bridge to cross a river to place a circle on the other side of the river, but PGA fails to do so.

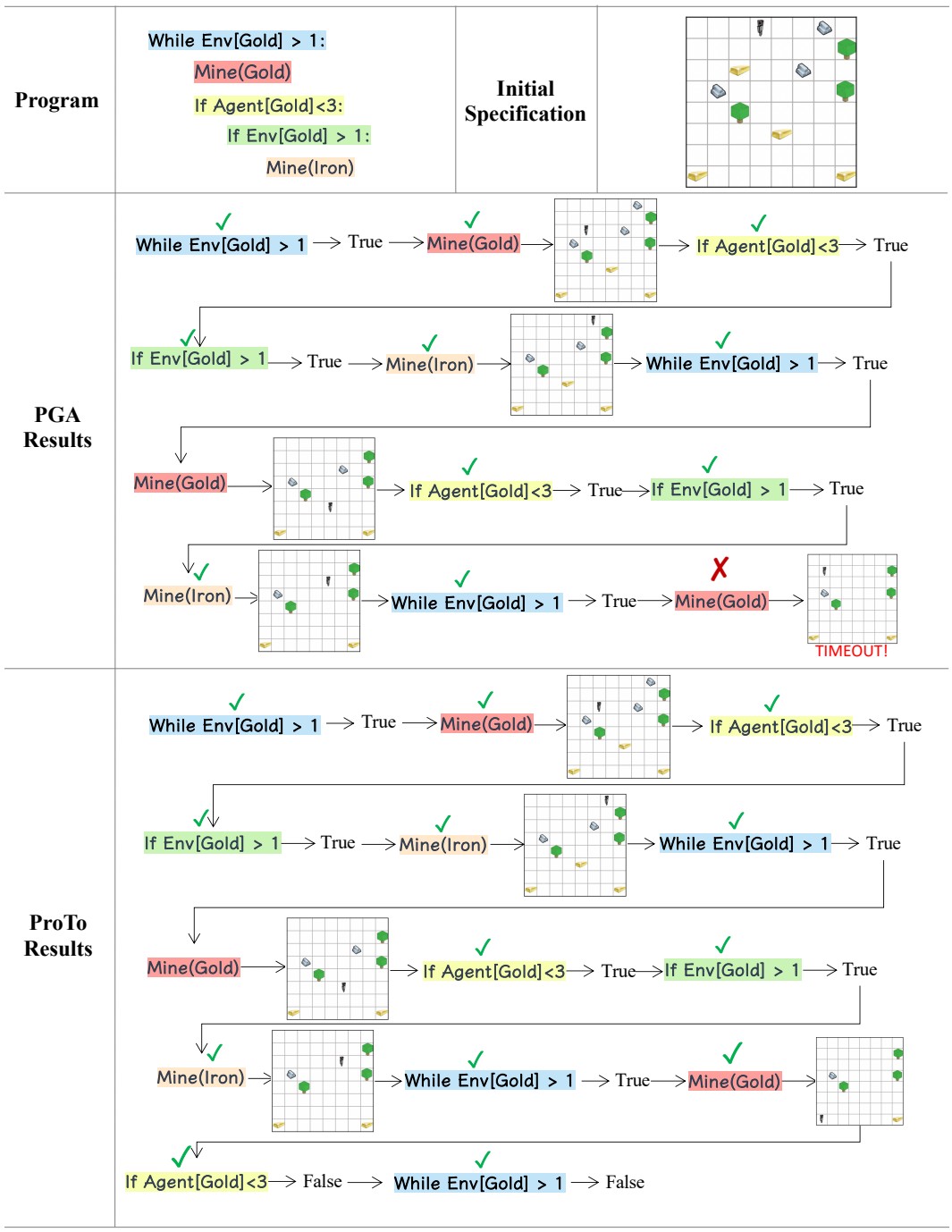

Figure 4: Visualization of test results on the Minecraft environment in comparison to program-guided agent [81] (Part 2). Due to space limitations, we omit the middle steps for executing routines and directly show the execution results of routines. We use the green tick and the red cross to denote right and wrong reasoning results, respectively. We observe that the ProTo agent can mine the gold in the corner, but PGA fails to do so within the time constraint.

# A    Extended Related Work

**Transformers with Masked Attention**    We notice a few transformer-based architectures adopt masked attention to achieve different goals. First, the masked self-attention [87] is adopted to restrict the model from seeing subsequent positions of tokens for machine translation. Second, mask attention networks [28] uses mask matrices to enforce the localness modeling ability of transformers. Third, in the vision domain, attention masks are used to highlight specific classes of visual content (e.g., foreground objects) [78, 91]. One concurrent unpublished work [34] uses masked attention to model data flow relationship for source code summarization.

**Neural Symbolic Learning**    The goal of neural symbolic learning is to pursue a coherent, unified view of symbolic logic-based computation and neural computation [9, 35, 36]. Previous neural symbolic systems involve visual reasoning [99, 64], logic induction [63], and reading comprehension [19]. We propose to leverage transformer architecture to integrate perception, reasoning, and decision. We believe transformer architecture would advance neural-symbolic systems because of its ability of compositional representation learning.

# B    Program Details

In this section, we provide more details of the programs on two experimented tasks.

## B.1    Program-guided Visual Reasoning

All types of routines on the GQA dataset [47] are provided in Table 8. No loop or branching routines exist in the GQA datasets. In the GQA dataset, the entrance routine is the first routine in the program, and the exiting routine is the last routine of the program, which would produce the final answer to the program. Some types of routines may use more than one inputs.

There are three types of program inputs and outputs (execution results). The first type is `Objects`, which is a probabilistic distribution over detected objects [64]. The second type is `Boolean` that is either True or False. Finally, the `Answer` type is a distribution over all answer candidates. We acquire the ground true execution results by executing ground true programs on the ground true scene graphs. Since the dimensions of results for different types of routines are different, we use MLPs with different output dimensions to decode the result embeddings of different routines. Note that ProTo executes the program in a latent space, so no explicit inputs are directly sent to ProTo. Only latent embeddings are sent into ProTo. The latent result embeddings are grounded to the explicit results via execution losses.

All the routines on the GQA datasets are single-step routines, which are finished via only a single forward step. In other words, we always update the pointer at each execution step.

## B.2    Program-guided Policy Learning

We list different types of routines on the 2D Minecraft datasets in Table 5. In this dataset, the entrance routine is the first routine in the program, and the exiting routines are routines that might be executed at the end of the program execution [4].

All routines in 2D Minecraft datasets take the specification as input and output either actions or Boolean results. The actions can either be motor actions or interactive actions as described in Sec 5.2 of the main text. Like the GQA experiments, we also use MLPs with different output dimensions to decode the results embeddings.

The list of ending conditions is also provided in Table 5. Note that the PGA baseline [81] uses the same set of end conditions.

We visualize the distribution of program lengths on both GQA and Minecraft in Figure D6.

---

[4]In branching cases, the last routines of both branches are possibly executed, so they are all exiting routines.

## C  Algorithm Details

### C.1  Derive Parents of Routines

For the GQA dataset, the parents of routines are provided in the ground truths. For example, in the Figure 2 of main text, the parents of the third routine (`Verify_relation(left)`) are the first routine (`Select(bag)`) and the second routine (`Select(wine)`).

For the Minecraft dataset, the parent of one routine is the previously executed routine. Each routine only depends on the previous routine, and no routines rely on more than one routine.

### C.2  Parallel Execution

Since ProTo is a transformer-like [87] architecture, it has a promising ability to execute many routines in parallel when they have no result dependency. A typical example is shown in Figure C5. The parallel version of algorithms is shown in Algorithm 3 and Algorithm 4. We adopt a vector of pointers $\mathbf{p}$ to point to different routines executed in parallel.

The semantic guidance remains the same as Eq. 1. We revise the structure guidance $\mathbf{P}$ to support executing many routines in one time as the following equation:

$$\mathbf{P}^{t(\tau)}[i][j] = \begin{cases} 0 & \text{if } (\exists p \in \mathbf{p}, \text{ s.t. } P_j \in \text{Parents}(P_p) \text{ and } P_i = P_p) \text{ or } (i = j), \\ -\infty & \text{else.} \end{cases} \quad (6)$$

Parallel execution is only enabled in the GQA experiments. On the Minecraft datasets, there is only one agent who needs to perform all the required tasks. So the routines in the Minecraft experiments can not be executed in parallel. We expect this parallel execution feature can be tested on a multi-agent environment [57] in the future.

Note that in the main text, we explain our approach sequentially for ease of understanding.

---

**Algorithm 3:** ProTo Execution in Parallel

**Result:** Execution results $\{o^{(\tau)}\}_{\tau=1}^T$
1. Initialize $\tau = 0$ and a pointer;
2. Build $\mathbf{P}^s$ according to Eq. 1;
3. **while** *not reach an exiting routine* **do**
4.     Observe $s^{(\tau)}$ and set $\tau = \tau + 1$ ;
5.     **if** *there are routines that can be executed in parallel* **then**
6.         Spawn a vector of pointers $\mathbf{p}$ to point to those routines;
7.     Build $\mathbf{P}^{t(\tau)}$ according to Eq. 6;
8.     $o^{(\tau)} = $ ProToInfer$(s^{(\tau)}, \mathbf{P}^s, \mathbf{P}^{t(\tau)}, \mathbf{p})$;
9.     Output $o^{(\tau)}$ ;
10.     ParaUpdatePointer$(\mathbf{p}, o^{(\tau)})$ ;
11. **end**

---

**Algorithm 4:** ParaUpdatePointer$(\mathbf{p}, o^{(\tau)})$

1. **if** *parallel routines finishes execution* **then**
2.     Merge pointers $\mathbf{p}$.
3. **for** *one pointer $p$ in $\mathbf{p}$* **do**
4.     **if** $P_p$ *does not finish execution* **then**
5.         **return**
6.     **if** $P_p$ *is an `If`-routine* **then**
7.         Point $p$ to the first routine in the $T/F$ branch if the result of $P_p$ is $T/F$;
8.     **else if** $P_p$ *is a `While`-routine* **then**
9.         Point $p$ to the first routine inside/outside the loop if the result of $P_p$ is $T/F$;
10.     **else if** $P_p$ *ends a loop* **then**
11.         Point $p$ to the loop condition routine.
12.     **else**
13.         Point $p$ to the subsequent routine;
14.     **end**
15. **end**

---

## D  Experimental Details and Further Results

In this section, we provide more details and further results on two experimented tasks.

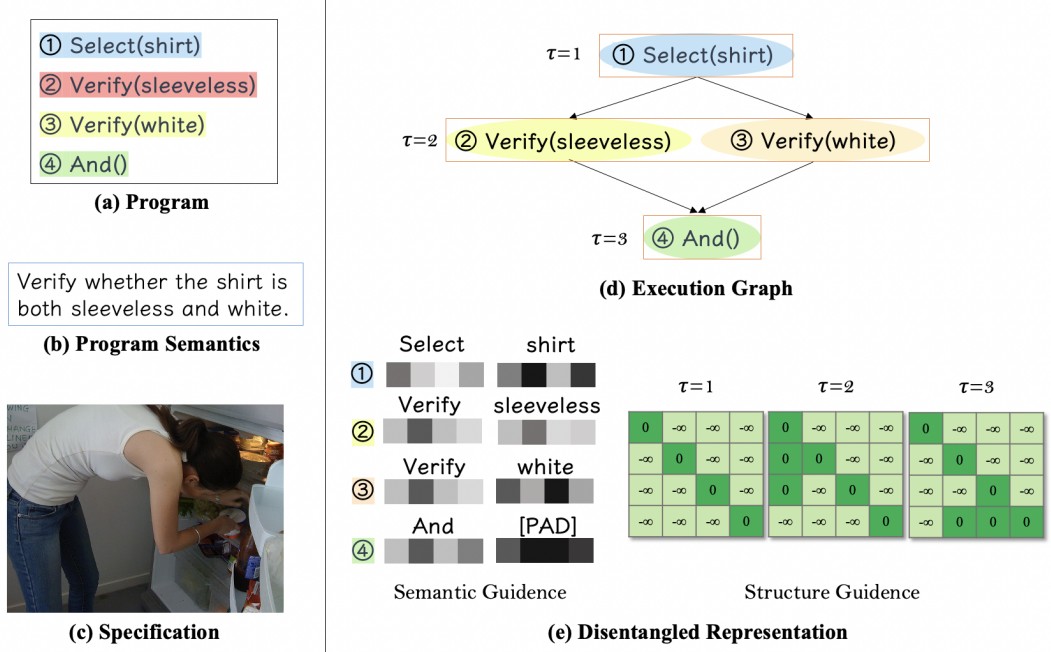

Figure C5: An example of ProTo parallel execution. We show the program and its semantics in (a) and (b). The specification is shown in (c). We show the execution flow in (d) where we can see we execute two routines at $\tau = 2$. In (e), we show the semantic guidance and the structure guidance of ProTo execution process.

## D.1 Program-guided Visual Reasoning

### D.1.1 Program Synthesis Model

We adopt a simple transformer-based seq2seq model [87] to translate a natural language question into a program. Both the encoder and the decoder of the seq2seq model are composed of six identical self-attention layers with hidden feature dimensions $d_h = 512$. The head number is eight.

The input question is encoded in a token-by-token manner via a learnable dictionary $\phi_q$. We turn the ground true program into a sequence by traversing the program tree via pre-order traverse. Segment tokens [SEG] are added between two routines. The predicted sequential program can be recovered via a reverse way. We used beam search with a beam size of 4 and length penalty $\alpha = 0.6$ [87]. The validation accuracy of this model is $98.1\%$.

### D.1.2 Program Representation

In the semantic part of the program, we set $d_m = 256$, $L = 8$ and $d = 2048$. In the implementation of the structure part, we use $-1 \times 10^9$ as the negative infinity, which is the same as the standard implementation of masked attention in transformers [87].

### D.1.3 Visualizations

We provide more visualization of ProTo on the validation datasets in comparison to the concurrent work meta module networks (MMN) [15]. The implementation and hyperparameters of meta module networks follow their official code. Specifically, for the Objects types of results, we visualize the predicted object with a probability $p > 0.5$. For the results with type Boolean or Answer, we present the choice with maximum probability. The visualizations are shown in Figure 1 and Figure 2.

### D.1.4 Details of the GAT Encoding

During the ablation study, we compare our model to graph attention networks (GAT) [88]. The node features are semantic embeddings of routines, and the edges represent message passing relationships.

Specifically, the node features are constructed via a concatenation of word embeddings, which is the same as Eq. 1. The embedding dimension is the same as ProTo. One edge exists between a routine and its parents as in Eq. 2. Note that since the GQA programs do not have conditional routines such as `While` and `If`, the edges are determined before execution. The routine embeddings are fed to two cross-attention modules to fuse information of result embeddings and specifications following Eq 3 and Eq 5. We also use an MLP to decode the latent results to get explicit routine results.

The GAT model consists of three layers, where each layer has eight attention heads with 256 features, following by an ELU nonlinearity.

### D.1.5 Details of Generalization Experiments

**Purpose of Collecting Additional Human-written Programs** We have the following reasons for collecting the human-written programs. First, we are curious whether humans can communicate with machines via programs, which has not been done by previous work before. Second, the GQA questions and programs are synthetic, and many of the programs are awkward (e.g., with many unnecessary modifiers such as "the baked good that is on the top of the plate that is on the left of the tray"). Third, the GQA test split programs are not publicly available, and the translated programs from the questions may be inaccurate. Since the validation split has been used for parameter tuning, we wish to benchmark program-guided visual reasoning on the collected independent data points. Forth, this small-scale dataset lays the ground for the construction of our novel dataset for program-guided tasks.

**GQA-Humam-Program Dataset Collection Process** For the Human Program Guidance experiments, we create the GQA-Humam-Program dataset to diagnose whether humans can guide the reasoning process via programs. We employ five volunteers to write 500 programs and answers on the GQA validation dataset. The estimated hourly wage is ten dollars, and the total amount spent on volunteers is two thousand dollars. A parser checks the written programs to ensure that they follow the domain specification language of GQA. We encourage the volunteers to write longer and more complex programs. Two volunteers cross-check the correctness of programs and answers. For fairness of comparison, we retrain the meta neural module networks [15] on the training split of GQA while preventing it from seeing the natural language questions. The screenshot of the data collection tool is provided in Figure D7.

**Rationale Behind Unseen Programs Experiments** In the experiments of Unseen Programs, the models are required to learn combinatorial word-level semantics to execute unseen programs. The training set contains `verify_size` and `filter_shape`, the models may generalize compositionally to the unseen program `verify_shape`.

**Restricted Data Regime Experiments** We repeat for three random seeds and found the standard deviation of the results is smaller than 2%.

### D.1.6 Computational Resources

We train our model and the baselines on a 48 core Ubuntu 16.04 Linux server with eight Nvidia Titan-X GPU. The CPU is Intel Silver 4116 CPU @ 2.10GHz. The total training time is around 48 hours.

### D.1.7 License and Permissions

The GQA dataset is built upon Visual Genome [53], which is under Creative Commons Attribution 4.0 International License. The GQA dataset is publicly available so that we can use it for research purposes. The GQA dataset is used in many published literature [47, 45], and we do not found offensive content in this dataset.

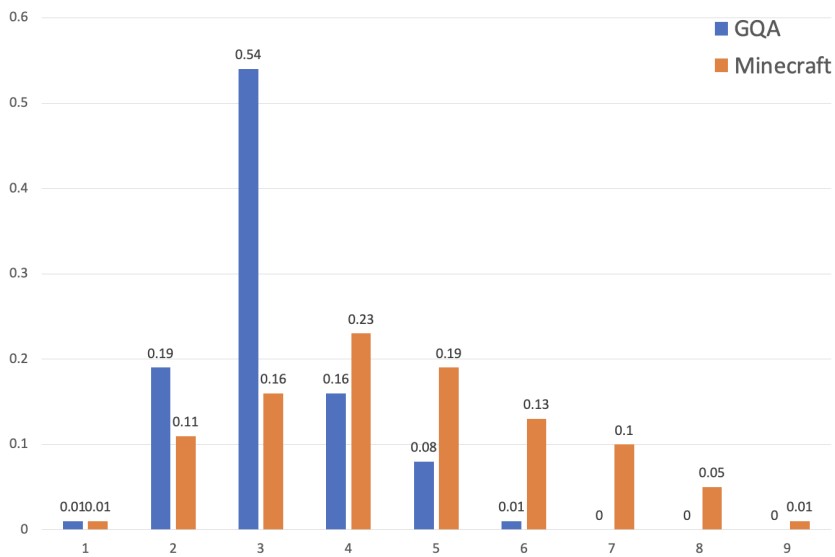

Figure D6: The program length distribution on the GQA and the Minecraft dataset.

## D.2 Program-guided Policy Learning

### D.2.1 Environment Details

The major environmental resources that the agent can interact with are gold, wood, or iron. The environment may contain a river, and the agent cannot go across unless a bridge is built. The size of the grid world ranges from five to eight. There could be two to four merchants in the environment. The environment is randomly initialized, and the agent is also randomly initialized while ensuring the program can be finished. If the agent fails to finish the program within 300 timesteps, the execution would be terminated (timeout).

### D.2.2 A2C and Hyper-parameters

We use the same implementation of the A2C algorithm as [81] (provided by its authors). The A2C algorithm uses a learning rate of $1 \times 10^{-3}$, $64$ environments running in parallel with $64$ number of workers. The number of roll-out steps for each update is five. The agent is trained for $10^7$ timesteps. The balance of the entropy regularization term is $\beta = 0.1$.

### D.2.3 More Visualizations

We provide more visualizations of the results on the test splits in Figure 3 and Figure 4.

### D.2.4 Architecture Details and Computational Costs

We list the computational costs and details of ProTo and the baselines in Table 6.

The server that we used is the same as the GQA experiments, but we only use a single GPU in the Minecraft experiments following [81]. The total training time is around 80 hours.

### D.2.5 Details about the Planner Used in Dense Supervision

We create a planner to generate ground true execution traces to train the ProTo baselines with full supervision. The planner uses a hard-coded interpreter to parse the programs. For all the actions that need navigation, the planner uses an A* search algorithm [26] to find the shortest path. For the Bridge, Mine and Sell action, we would find the nearest river, the nearest item or the nearest merchant. For the If and While routines, the planner uses the symbolic information in the environment (e.g. env[gold]=3) to decide whether the conditions are satisfied.

GQA-Humam-Program Construction

Please give a program about the following image and provide the correct answer to the program. Note:
(1) The program should follow GQA domain-specific language.
(2) The answer should be in GQA answer vocabulary.
(3) The program should be diverse and complex with a length of 3-10.

1.

Provide a program about the above image. *

| Routine Type | Arguments |
| --- | --- |
|  |  |
|  |  |

+

2.Give the answer to the program. *

3.Rate the complexity of the program. *

○ 1      ○ 2      ○ 3      ○ 4      ○ 5
low                                      high

Figure D7: Screenshot of the UI for the construction of the GQA-Humam-Program dataset. The submitted programs are checked by the parser and the collected answers are cross-checked by two more annotators.

### D.2.6 License and Permissions

The Minecraft dataset is under Creative Commons Attribution 4.0 International License. We acquire this dataset and its license from the authors of [81]. Since it's a synthetic dataset, we don't think it has offensive content.

## E    Additional Discussions about the Neural-Symbolic Baselines

The comparison between different neural symbolic approaches is a significant aspect of our work. On the first domain of visual reasoning, our paper shows that a mixed parametric neural-symbolic model would outperform pure symbolic non-parametric executors (Table 2). On the second domain of

Table 5: All types of routines in the domain-specific language of the 2D MineCraft dataset [81]. We use Spec. to denote specification. Some routines require many arguments, which are numbered by integers with brackets (e.g., (1), (2), etc.).

| Type | Arguments | Input | Output | Semantics and Ending Conditions |
|------|-----------|-------|--------|----------------------------------|
| Mine | Triangle, circle, rectangle, gold, wood, or iron | Spec. | Action | Go to the item and pick it up. Ended when the action "Mine" is predicted and the agent successfully mined a gold. |
| BuildBridge | - | Spec. | Action | Go to the river and build a bridge. Ended when the action "Bridge" is predicted and the agent successfully builds a bridge. |
| Goto | Coordinates | Spec. | Action | Go to the coordinates. Ended when the agent reaches the target. |
| Place | (1) Triangle, circle or rectangle (2) Coordinates | Spec. | Action | Place the object on the specified coordinates. Ended when the action "Place" is predicted. |
| Sell | Triangle, circle, rectangle, gold, wood, or iron | Spec. | Action | Go to the merchant and sell mined items. Ended when the action "Sell" is predicted. |
| If / If-Else | (1) Agent, env, or is_there (2) Gold, wood, iron, bridge, river, merchant, wall, or flat (3) Operators $(>, \geq, =, <, \leq)$ (4) Number | Spec. | Boolean | Perform branching based on the condition in the if-clause. Ended in a single execution step. |
| While | (1) Agent, env, or is_there (2) Gold, wood, iron, bridge, river, merchant, wall, or flat (3) Operators $(>, \geq, =, <, \leq)$ (4) Number | Spec. | Boolean | Perform looping based on the condition in the while-clause. Ended in a single execution step. |

Minecraft, we have compared neural baselines (Table 4). We also have a pure symbolic (non-learning) method: the planner. The advantages of the learned executor beyond the symbolic planner are as follows.

First, the hard-coded planner requires a lot of ad-hoc engineering work to handle complex cases. But our method can learn from a sparse reward signal. For example, our method can learn to build a bridge to cross the river to fetch gold on the other side of the river without explicit supervision (just given a reward signal). However, a planner needs to handle this case with special treatment.

Second, we experiment on Minecraft to validate the ability of ProTo to generalize across unseen routines. Specifically, we remove a routine from the training split (e.g., Mine(Gold)) and test on programs that contain the removed routine. This experiment validates the compositional generalization ability (e.g., generalize to Mine(Gold) after seeing Is_there(gold) and Mine(Silver)). Only a

Table 6: Architecture details of ProTo and the baselines on the 2D MineCraft dataset.

| Model | Parameters | Architecture Details |
|---|---|---|
| ProTo | 1.30M | Eight attention heads with an intermediate size of 64; shared computation between routines; the state map is encoded via a two-layer MLP with hidden size 256 and output size $d = 2048$; the inventory is encoded via another two-layer MLP with hidden size 128 and output size $d = 2048$; the inventory feature is added to the state map to produce the final specification feature; the output MLP is also a two-layer MLP with hidden size 256. |
| PGA | 1.21M | The state map is encoded via a batch of CNNs with channel sizes of 32, 64, 96, and 128. Each convolutional layer has kernel size three and stride 2, which is followed by ReLU nonlinearity. The inventory is encoded via a two-layer MLP with a channel size of 256. The goal is encoded via a two-layer MLP with a channel size of 64. The features are fused via a modulation mechanism proposed by PGA [81]. |
| Vanilla Transformer | 2.63M | Two attentional layers stacked, with eight attention heads, a hidden size of 128, and an intermediate size of 256. |
| Tree-RNN | 0.51M | Program embeddings are of dimension 128. Attention LSTM size of 128. Tree-RNN uses a composition module to aggregate all the children representation of a node, which is of size [128 × 128], and output projection weights of size [128 × 128], with a bias of size 128. The program embeddings are average pooled across one routine so that each routine will be mapped to a fixed dimension. The composition layer is applied when combining pooled embedding from all the children of a node. |

reward signal is used for training. Other experimental details are the same as described in Sec 5.2. The planner is set to choose a random legal action when meeting an unseen routine). The results on the validation dataset are presented in Tab 7.

Third, the planner cannot scale up to a large number of states. A planner cannot work well on a large-scale scenario such as the game GO [76].

Forth, the planner can never work on raw image observations. Note our transformer-based architecture can work on raw image input after dividing the observation into patches [25] or detecting objects in the raw image. But the symbolic planner has no way to work on raw image inputs.

## F   Limitations

Despite our contributions to task formulation and ProTo models, our work has several limitations. First, since the program-based approaches need to leverage the program guidance and dense supervision, it cannot easily leverage large-scale datasets for self-supervised pretraining [61]. We would work on building large-scale datasets with program annotations to alleviate this limitation. Second,

Table 7: Comparison between the generalization ability of the planner and ProTo on the Minecraft dataset.

| Removed Routine | Planner Acc | ProTo Acc |
|---|---|---|
| Mine(Gold) | 11.4 | 59.3 |
| Is_there(River) | 43.8 | 65.6 |
| Agent[Silver] | 46.2 | 77.1 |

we hypothesize that one can still improve proTo's architecture design since the community has not exploited the power of transformers. We would incorporate recent advances in transformers [59, 100] to improve ProTo. Furthermore, we could conduct more ablation studies to reveal the importance of different components in ProTo.

## G   Broader Impact

Our findings provide a simple yet effective approach to address program-guided tasks. Potentially, people may leverage ProTo models to instruct robots via programs [69]. However, the learned executors may be attacked by adversarial training [62]. And instructing machines via programs might require humans to have more advanced knowledge (e.g., knowing the basic concepts of programs and how to follow program syntax). Therefore, those of a low educational level may not be able to leverage the benefits of our approach, which might aggravate social inequalities.

Table 8: All types of routines in the domain-specific language of the GQA dataset [47].

| Type | Arguments | Input | Output | Semantics |
|---|---|---|---|---|
| Select/Filter | Position, color, material, shape, or activity | Objects | Objects | Filter out a set of objects by the positions, colors, etc from the input objects. |
| Choose | Name, scene, color, shape, position or attributes | Objects | Answer | Choose one answer (e.g., name, scene, color) from given answer candidates. |
| Verify | Color, shape, scene, or relation | Objects | Boolean | Verify whether the given concepts (e.g., color, shape) holds true for the input objects. |
| Relate InverseRelate | Name, or attributes | Objects | Objects | Filter out a set of objects that have the relation concept (e.g., names, attributes) with the input objects. |
| Query | Name, color, shape, scene or position | Objects | Answer | Query the concept (e.g., name, color) of the input objects. |
| Common | Color, or material | Two objects | Answer | Query the common concepts (e.g., color, material) of the input objects. |
| Different | Name, color, or material | Two objects | Boolean | Return whether the concepts of the objects (e.g., name, color) are different. |
| Same | Name or color | Two objects | Boolean | Return whether the concepts of the objects (e.g., name, color) are same. |
| And | - | Two booleans | Boolean | Return whether the two input booleans are both True. |
| Or | - | Two booleans | Boolean | Return whether one of the two input conditions is True. |
| Exist | - | Objects | Boolean | Return whether the input object set is not empty. |