# OpenReview forum: "ProTo: Program-Guided Transformer for Program-Guided Tasks"
_NeurIPS.cc/2021/Conference — NeurIPS 2021 Poster_

### Official Review · Reviewer_5PzE · 2021-06-26

**Rating:** 6
**Confidence:** 4

**Summary:**

This paper explores how learning agents can follow program guidance to solve visual reasoning and policy learning tasks. While most prior works are only designed to address one of the two tasks, this paper aims to propose a unified framework that can tackle both two problems. To this end, the paper proposes a Transformer-based framework (ProTo) that learns to interpret and execute programs on observed specifications by learning to execute a program in a latent space with cross attention and masked attention mechanisms. The experiments on the visual reasoning task (GQA dataset) show that ProTo outperforms baselines using different learning signals such as question-answer pairs, scene graphs, and programs. The experiments on the program-guided policy learning in a 2D-Minecraft environment show that ProTo outperforms baselines and generalizes better to long and complex programs. I believe this work studies a promising research direction and proposes a framework that can address program-guided tasks in two domains. Therefore, I am leaning toward accepting this paper yet still have some concerns (see below) which I hope to be addressed in the rebuttal.

**Ethical Concerns:**

I currently do not recognize any potential negative impact or ethical concerns for this work.

**Limitations And Societal Impact:**

Described in the main review section.

**Main Review:**

## Paper strengths and contributions
**Motivation**
- The motivation for using programs as a structured, clean and formal representation to guide learning agents for task-solving is convincing.

**Novelty and technical contributions**
- To the best of my knowledge, the idea of unifying two program-guided task domains, including visual reasoning and policy learning, is novel.
- The idea of separately leveraging the program semantics and the explicit structure of the given program is convincing, and this paper proposes a way to implement this idea.
- Treating routines (sub-tasks in programs) as objects and leverages attention mechanisms to model interactions among routines and specifications using Transformer seems effective.
- Execute programs in a learned latent space for strong representation ability seems to contribute to the performance gain and generalization capability.
- The proposed framework can be trained using different levels of supervisions, including dense execution traces, a goal specification with distance measurement, and sparse rewards. This provides great flexibility.

**Clarity**
The writing of the introduction and the related work is clear. The problem and the proposed framework are well motivated and the contributions are explicitly stated.

**Experimental results**
- The presentation of the experimental results is clear.
- Visual reasoning task on the GQA dataset:
	- ProTo outperforms all baselines using learning signals including question-answer pairs, scene graphs, and programs on all the metrics.
	- The generalization experiments evaluate how well the models can generalize to the programs that are written by humans. The results show that ProTo outperforms the meta module network.
	- The low data regime experiment evaluates if how well the models can perform with only 10% of training data. The result shows that ProTo outperforms the meta module network.
- Policy learning task on the 2D-Minecraft domain:
	- ProTo learns more efficiently compared to the program-guided agent baseline and end-to-end learning models.
	- The experiments on longer and more complex programs show the better generalization performance of ProTo.

**Ablation study**
The ablation studies on the GQA dataset show that:
- Leveraging both the structure guidance and the semantic guidance is the key to the superior performance of ProTo.
- Utilizing a Transformer to encode inputs is better than using a graph neural network.

**Reproducibility**
The code is provided, which helps understand the details of the proposed framework.

## Paper weaknesses & questions

**Clarity: method section**
The writing of the method section is unclear and is not easy to follow. I have read it three times and still am not sure I correctly understand what modules are learned to do what, what input/output representations they use, how they are used for tasks from two domains. With my partial understanding of the proposed method, it is not easy for me to judge it. I believe a significant revision of the method section is required.

**Walk-through example or flow chart**
Together with the point mentioned above, I believe it would be better to either provide a walk-through example or a flow chart that explicitly shows the learning modules and the input/output representations. It would be even better if walk-through examples from both domains are shown. While Figure 2 is supposed to serve this purpose, it is not still clear to me.

**Table 2 bottom rows**
I do not know how to interpret the bottom two rows of Table 2. Do the numbers indicate the performance of NS-VQA and MMN with partial supervision? If this is the case, I suggest the authors move the midrule a row higher so that three models (ProTo, NS-VQA, and MMN) learning from partial supervision and put them together for comparison. If not, the authors should probably provide a better description at the end of Section 5.1.

**Unseen programs**
I do not understand the unseen program generalization experiment on the GQA dataset. Does it mean that the model has never seen verify type tokens during training and can still work when tested on a program with such tokens without further training? Also, why are the verify type of instances chosen not others? It would be better if the authors offer intuitions for this.

**External memory**
While the authors believe that it would be better if a model does not leverage external memory for program-guided tasks for more efficient training,
I actually believe incorporating external memories would allow the framework to be applied to a wider scope of program-guided applications.

**Human-written programs and model-synthesized/GQA programs**
The generalization experiment evaluates how well ProTo and MMN can generalize to human-written programs, which are supposed to be different from model-synthesized programs and the programs in the GQA dataset. However, without some examples of all types of programs (human-written programs, model-synthesized programs, and GQA programs), it is not easy to judge if the distribution gap exists or how large the gap is, making it even harder to judge the generalization ability of ProTo and MMN.

**Missing related works**
While the related work section is fairly sufficient, including the following works would make it more comprehensive:
- Neural Scene De-rendering
- Neural Program Synthesis from Diverse Demonstration Videos
- Learning to Describe Scenes with Programs
- Learning to Infer and Execute 3D Shape Programs

**Minor issues**
- duplicated citations: [57] & [58]
- L155: Note "that"
- Many other grammar errors

## Other metrics

### Relevance and significance
- Reasonable contribution to a minor problem

### Novelty
- One idea that surprised me by its originality, solid contributions otherwise

### Technical quality
- Technically strong, highly general results, advanced techniques

### Experimental evaluation
- Solid, informative evaluation w.r.t all 5 criteria

### Clarity
- This paper is difficult to understand in places because of typos, lack of organization, or other flaws. The paper would need significant/major improvement in writing.

**Time Spent Reviewing:**

11

---

> ### Author Response · Authors · 2021-08-10
> **Reply to R4 5PzE: thank you!**
>
> We thank your positive and detailed reviews! We are encouraged by your appreciation for the motivation, novelty, and clarity of our work. We also appreciate your summary of the highlights in the experiment part.  We are happy to answer your questions as follows:
>
> ### Clarity of Method Section
>
> To understand the method part, please first look at Algorithm1 and Algorithm2. Basically, the algorithm executes the program in a routine by routine manner until the end of the program. There is only one learnable module (besides the learnable dictionaries), ProTo, visualized in the right of Figure 2. In other words, we do not learn ad-hoc modules for different routines but using one shared ProTo module for all routines. Different routines have different input types, but they use the same ProTo module. The forward process of ProTo corresponds to Line 6 of Algorithm1, represented as ProToInfer.
>
> The inputs to the model are the observation, the semantic and the structural part of the program, and the pointer. The model's output is a vector representing the execution results (e.g., distribution over answer candidates). One can find this part in the Line6 of Algorithm1 to see the input and output. We also state this in Sec 4.2, Line 152.
>
> Then we explain each of the inputs and outputs in the following table:
>
> | Name                            | I/o    | Notation               | Explanation                                                  |
> | ------------------------------- | ------ | ---------------------- | ------------------------------------------------------------ |
> | Specification                   | Input  | $s^{(\tau)}$           | The object-centric representation for the image or the map.  |
> | Semantic Part of the Program    | Input  | $\mathbf{P}^{s}$       | The semantic part representation for the program, described in Sec 4.1. |
> | Structure Part of the Program   | Input  | $\mathbf{P}^{t(\tau)}$ | The structure part of the program, which is updated at each execution timestep, is described in Sec 4.1. |
> | Pointer                         | Input  | p                      | The pointer is used to indicate the currently executing routine. |
> | The output of Executed Routines | Output | $o^{(\tau)}$           | The distribution over the set of output candidates (e.g., an answer or an action). The types of routine outputs are shown in Table 5 and Table 7 in the Appendix. |
>
> We hope the above explanation can help you to understand our methods. We will try our best to revise the method partly to make it clearer. Please feel free to raise questions if some parts still confuse you.
>
> ### Walk-through Examples
>
> We provide step-by-step examples in Figure 4 of the main text and Figure 1-4 of the Appendix (two for the GQA and two for the Minecraft). We will try to make Figure 2 clearer. If there is space in the main text, we will move examples into the main text.
>
> ### Table 2 Bottom Results
>
> Let us walk you through the whole Table 2. The first row is the full ProTo model's validation accuracy. The middle four rows are all ablated ProTo's accuracies. And the last two rows are other methods' validation accuracies for reference. The Partial Supervision (the third-to-last row) is an ablated version of ProTo as described in Line 236-238. Both the NS-VQA and MMN are trained with dense supervision. We would highlight this at the end of Sec 5.1. Thanks!
>
> ### Unseen Programs
>
> The type of a program is determined by the type of the last routine in the program. So when we remove the verify-type of programs, the last routines cannot be verify-type. But there still can be verify-type tokens in other places of routines. For example, the program related to the logical question "Is the fruit on top of the tray yellow and thick?" requires two routines of ”verify(thick)“ and “verify(yellow)”. Note this question is not verify type and is not removed from the training splits. Please also note the general response for an explanation on this.
>
> We choose the verify-type for validation because we follow the validation experiments of the MMN baseline [1].
>
> ### External Memory
>
> We agree with your opinion that external memory can potentially help extend the scope of ProTo in the future. Our claim of "removing memory helps efficient training" should base on the current datasets. We would make it more rigorous in the next version of the paper.
>
> We observe a trend that transformer-based architecture outperforms RNN-style architecture, where the latter uses a recurrent memory mechanism (such as LSTM). Furthermore, we feel difficult to conduct batched training if each data point relies on external memory. Besides, we found the gradients in the memory would vanish over time if the external memory is large. We hope future work can investigate this and give better understandings.
>
> ### Human-written Programs
>
> In GQA experiments, the model synthesized programs are almost equal to the GQA programs because of the effectiveness of the program synthesizer (mentioned at the bottom of Page 6).
>
> To help you understand the differences between our collected programs and GQA programs (acquired by the trained program synthesizer) on the test dataset, we calculate the following statistics:
>
> 1. **Match rate** = # matched programs / # all programs. If a program in one dataset can be found exactly in another dataset, then we call this a matched program.
> 2. **The average length of programs**. This measures the complexity of the collected programs.
> 3. **The average number of filter routines in one program**. Some GQA programs involve many unnecessary 'filter' operations. For the example,  "the baked good that is on the top of the plate that is on the left of the tray", the program would filter the tray, then the plate, and the baked good.
> 4. **The ratio of complex programs**. We define a routine to be complex if it takes two objects or boolean as inputs (e.g. Common, Different). A program is defined to be a complex program if a complex routine is in the program.
>
> The results are as follows:
>
> | Metric                    | Official GQA | Collected GQA-Human-program |
> | ------------------------- | ------------ | --------------------------- |
> | Match rate                | 13.4%        | 20.1%                       |
> | Avg. length               | 3.1          | 4.6                         |
> | Avg. # filter-routines    | 1.9          | 0.6                         |
> | Ratio of complex programs | 15.2%        | 33.0%                       |
>
> The results show that few programs can be exactly matched between GQA and our collected dataset. As for the complexity, GQA-HUMAN-PROGAM is more complicated with longer and more complex programs. Note in the collection process, we encourage the annotators to make complex programs (see Appendix Figure D6). We found that the GQA-HUMAN-PROGRAM contains fewer filter routines because filter routines are often unnecessary (many of the GQA images contain one instance, so filter routines are not needed).
>
> ### Missing Related Work & Typos
>
> Thanks for your suggestions! We have modified the typos and added the missing related work in our draft. Note "Learning to Infer and Execute 3D Shape Programs" was already referred in the original submission (Line 60, ref [77]).
>
> Please feel free to ask us if you have more questions!
>
> **References**
>
> [1] Chen, Wenhu, et al. "Meta module network for compositional visual reasoning." *Proceedings of the IEEE/CVF Winter Conference on Applications of Computer Vision*. 2021.

---

> > ### Comment · Reviewer_5PzE · 2021-08-13
> > **Re: Reply to R4 5PzE: thank you!**
> >
> > Thanks for the detailed response.
> >
> > Many of my questions have been properly addressed, including clarity of the method section, walk-through example or flow chart, Table 2 bottom rows, missing related works, and minor issues.
> >
> > **Unseen programs**: if this (what the authors describe in the response) is the case, the statement in the paper "we remove from the training split all the verify type of instances" seems misleading. I suggest the authors revise the description of the generalization experiment.
> >
> > **External memory**: the authors should explicitly state that external memory might not be effective for the tasks considered in this paper. Also, just to be on the same page, I would consider RNNs' internal state as internal memory, not external memory discussed here. By external memory, I meant explicit memories such as processor registers or differentiable memories in neural Turing machines.
> >
> > **Human-written programs**: I appreciate the additional information provided here. I would still like to see some randomly sampled example programs.

---

> > > ### Author Response · Authors · 2021-08-16
> > > **Reply to your further questions**
> > >
> > > Thanks for your quick and thoughtful reply to our response, and we feel glad that our response addressed many of your questions.
> > >
> > > ### Unseen Programs
> > >
> > > We realize that the term used in our original submission, "verify type of instances," is confusing. We have revised it after reading your helpful reviews. Thanks!
> > >
> > > ### External Memory
> > >
> > > Sure, we would explicitly state that the external memory is not effective in our exemplar tasks. How to make differentiable and efficient external memory is still an open problem in program-guided tasks. We encourage more work to explore this.
> > >
> > > ### Human-written Programs
> > >
> > > We randomly sample some human-written programs, which are provided in the following anonymous link:
> > >
> > > https://anonymous.4open.science/r/neurips_visualization-2EEE.

---

> > > > ### Comment · Reviewer_5PzE · 2021-08-16
> > > > **Re: Reply to your further questions**
> > > >
> > > > Thank you for the response. I have no further questions at this moment.

---

### Official Review · Reviewer_JPZw · 2021-07-14

**Rating:** 6
**Confidence:** 4

**Summary:**

This work proposes Program-Guided Transformer (ProTo) for program execution tasks. Specifically, instead of simply encoding the program as a token sequence, they disentangle the program representation into a semantic representation and a structural representation, and the structural representation takes the program control flows into accounts. They modify the attention mechanisms of the vanilla Transformer architecture to encode the disentangled program representation. They evaluate their approach on two tasks: (1) program-guided visual question answering on GQA; and (2) program-guided policy learning task proposed in [64]. They show that ProTo outperforms other baselines.

**Limitations And Societal Impact:**

The authors adequately addressed the limitations and potential negative societal impact.

**Main Review:**

Learning to execute programs is an important topic. In particular, executing programs on visual inputs and understanding routines in programs are challenging problems. For both GQA and policy learning tasks, the authors present good results and comparisons to strong baselines. The proposed Transformer architecture properly utilizes the program structure. Although the programs evaluated in this work are in restricted domain-specific languages, the ProTo architecture might be extended to more complicated programming languages.

I have a couple of questions about the approach design and experiments:

1. From Algorithms 1 and 2, the pointer update depends on the execution results. According to the equation for computing the output o, I suppose the execution output vector should represent a probability distribution over all possible execution results. When you train or evaluate the model, do you take the argmax to determine the execution output for the pointer update, e.g., selecting T/F for conditional statements and selecting actions for executing routines? If this is the case, is the model hard to train since wrong intermediate execution results can mislead the execution of all remaining program statements?

2. Could you present the distribution of program lengths in your experiments? What is the largest program length ProTo can handle? From the attention computation and the structural representation, the transition mask and some intermediate results are |P|x|P|-dimensional, thus the model might not scale to larger programs without modification.

3. A general suggestion on related work and baseline discussion: I feel that the main point of this work is about neural-symbolic reasoning and program understanding, because the modification of the Transformer architecture itself is not extremely novel. Therefore, in the main body of the paper, the authors can include more discussion of existing work on neural-symbolic learning, and the key differences between this work and baseline models that lead to better performance.

4. I am confused about the results of Unseen Programs in GQA experiments. For removing "verify"-type problems, do you mean that you only remove them for training the ProTo model, or do you also remove them for training the program synthesis model? If you completely remove all these problems for training, I don't know why the model can generalize. Specifically, how should the model understand these new functions if they never appear in training?

5. What is the point of collecting additional human-written programs for evaluation? Why not simply evaluate the model performance on ground truth programs in GQA? Could you provide the statistics of the human program dataset you collect, and explain what are the key differences between these new programs and those in GQA?

6. In the policy learning experiment, it says that "Transformer and TreeRNN encode the input program in a token by token manner". However, I suppose the TreeRNN model should encode the program as a tree. Could you explain more about how you use TreeRNN here?

7. In experiments of Longer Programs and Complex Programs, do you only train the models on shorter programs with fewer control flow constructs, or do you still train on programs of all different complexities, but present the test results on harder programs? I wonder whether these results show the generalization of models to programs that are more complex than training samples, or they are more about breakdown results varying the program complexities.

**Time Spent Reviewing:**

2

---

> ### Author Response · Authors · 2021-08-10
> **Reply to R3 JPZw: thank you!**
>
> We thank your encouraging, deep and thoughtful review! We are delighted that your appreciation of the importance of learning to execute. We also think routines are worth more exploration in future work as they are the basic elements of programs. We also hope that our framework can be extended to large-scale complicated Turing-complete languages, which would start a new era of neural-symbolic AI.
>
> We answer your questions as follows:
>
> ### 1. "Argmax Issue" of Branching Cases
>
> Booleans are treated as actions in our framework as described in Line 274, so the sampling process of the distribution is the same as other actions. In experiments, we use a Categorial sampler function (but not a naive greedy sampler "argmax") that is commonly used in actor-critic RL approaches.
>
> For your second question, "is the model hard to train since wrong intermediate execution results can mislead the execution of all remaining program statements", I'm assuming you are referring to the RL Target setting (because we can easily train the condition routines under supervised learning).  This is quite a good question, and I'd like to answer you by the following folds:
>
> 1.  A wrong choice of the boolean action would indeed lead to a wrong trace. But it's a common unstable issue in RL: a wrong decision in an MDP would often lead to a wrong direction. For example, in the game GO, slightly different places in the board would correspond to dramatically different chances of winning. An RL algorithm can successfully deal with this problem [1].
> 2. Since a random guess of the boolean action would have a 50% chance to get it correct, the exploration is not hard. In the StarCraft game [2], the action space is very large: hundreds of different units and buildings need to be controlled at once, in real-time. So we believe a SOTA RL algorithm is capable of addressing this problem.
> 3. Empirically, we do not observe a severe effect caused by this concern under our Minecraft experiments.
> 4. This also involves the long-horizon problem, which is also interested by R2. We are looking forward to future work in this direction.
>
> ### 2. Program Length Distribution and Computational Costs
>
> We present the program length distribution as follows. The largest program lengths are 6 and 9 for GQA and Minecraft, accordingly.
>
> | Length | Ratio on GQA | Ratio on Minecraft |      |
> | ------ | ------------ | ------------------ | ---- |
> | 1      | 0.01         | 0.01               |      |
> | 2      | 0.19         | 0.11               |      |
> | 3      | 0.54         | 0.16               |      |
> | 4      | 0.16         | 0.23               |      |
> | 5      | 0.08         | 0.19               |      |
> | 6      | 0.01         | 0.13               |      |
> | 7      | 0            | 0.10               |      |
> | 8      | 0            | 0.05               |      |
> | 9      | 0            | 0.01               |      |
>
> As for the computation issue, note the original transformer paper [3] includes the computation of mask for machine translation, which does not seem like a severe issue in the current transformer research. The result embedding matrix is essential for result passing and loss calculation in program-guided tasks. In practice, we found our model can be well fit into eight Titan X cards on GQA experiments. The training time is also acceptable (~48 hours). Speeding up ProTo is also an interesting and important topic.
>
> ### 3. Suggestions on Related Work
>
> Thanks! We would add more discussion on the neural-symbolic approaches. R1 and R2 gave some interesting axes on this, which would strengthen our claims.
>
> ### 4. Unseen Programs in the GQA dataset
>
> Sorry for the confusion! The program synthesis model is the same for all the validation experiments. We do not change the program synthesis part because we focus on program-guided tasks. So the verify-types of programs are only removed for training the ProTo model.
>
> In GQA, the type of a program is determined by the last routine of the program. Although the verify-type of programs are removed from the training split, the model still has a chance to see the "verify" routine in the training split (as long as it is not the last routine). For example, the program related to the logical question "Is the fruit on top of the tray yellow and thick?" requires two routines of" verify(thick) "and "verify(yellow)". Note that this question is not verify-type (but is an and-type logical program) and is not removed from the training splits. We would illustrate this in the next version of the paper. Please also note the general response for an explanation on this.
>
> ### 5. Purpose of Collecting Additional Human-written Programs
>
> Thanks for asking! We have the following reasons for collecting the human-written programs:
>
> 1. We are curious whether humans can communicate with machines via programs, which has not been done by previous work before.
> 2. The GQA questions & programs are synthetic and many of the programs are awkward (e.g., with many unnecessary modifiers such as "the baked good that is on the top of the plate that is on the left of the tray").
> 3. The programs on the GQA test split are not publicly available, and the translated programs from the questions may be inaccurate. Since the validation split has been used for parameter tuning, we wish to benchmark program-guided visual reasoning on the collected independent data points.
> 4. This small-scale dataset lays the ground for the construction of our novel dataset for program-guided tasks.
>
> To help you understand the differences between our collected programs and GQA programs (acquired by the trained program synthesizer) on the test dataset, we calculate the following statistics:
>
> 1. **Match rate** = # matched programs / # all programs. If a program in one dataset can be found exactly in another dataset, then we call this a matched program.
> 2. **The average length of programs**. This measures the complexity of the collected programs.
> 3. **The average number of filter-routines in one program**. Some GQA programs involve many unnecessary 'filter' operations. For the previous example, "the baked good that is on the top of the plate that is on the left of the tray", the program would filter the tray, then the plate, and the baked good.
> 4. **The ratio of complex programs**. We define a routine as complex if it takes two objects or boolean as inputs (e.g., Common, different, etc.). A program is defined to be a complex program if a complex routine is in the program.
>
> The results are as follows:
>
> | Metric                    | Official GQA | Collected GQA-Human-program |
> | ------------------------- | ------------ | --------------------------- |
> | Match rate                | 13.4%        | 20.1%                       |
> | Avg. length               | 3.1          | 4.6                         |
> | Avg. # filter-routines    | 1.9          | 0.6                         |
> | Ratio of complex programs | 15.2%        | 33.0%                       |
>
> The results show that few programs can be exactly matched between GQA and our collected dataset. As for the complexity, GQA-HUMAN-PROGAM is more complicated with longer and more complex programs. Note in the collection process, we encourage the annotators to make complex programs (see Appendix Figure D6). We found that the GQA-HUMAN-PROGRAM contains fewer filter routines because filter routines are often unnecessary (many of the GQA images contain one instance, so filter routines are not needed).
>
> ### 6. Tree RNN Baseline Information
>
> TreeRNN [5] is a baseline introduced by the previous work PGA [4]. The implementation and details of TreeRNN are from PGA's official code. The basic idea of TreeRNN is to summarize the fathers' embeddings and pass them to their sons via a recurrent mechanism. We refer you to PGA's Appendix E.4.2 for more details (we did not change the TreeRNN baseline).
>
> ### 7. Settings about Longer and Complex Programs
>
> The model is trained on standard (short) programs. Note in Line 299, "we test the trained agent in different settings.", which means we do not re-train the agent on the long and complex programs but test the trained agents on different generalization scenarios. So they are showing the generalization of models to more complex programs.
>
> We look forward to discussing with you if you have future questions!
>
> **References:**
>
> [1] Silver, David, et al. "Mastering the game of go without human knowledge." *nature* 550.7676 (2017): 354-359.
>
> [2] Vinyals, Oriol, et al. "Grandmaster level in StarCraft II using multi-agent reinforcement learning." *Nature* 575.7782 (2019): 350-354.
>
> [3] Vaswani, Ashish, et al. "Attention is all you need." *Advances in neural information processing systems*. 2017.
>
> [4] Sun, Shao-Hua, Te-Lin Wu, and Joseph J. Lim. "Program guided agent." *International Conference on Learning Representations*. 2019.
>
> [5] Alon, Uri, et al. "code2seq: Generating sequences from structured representations of code." *arXiv preprint arXiv:1808.01400* (2018).

---

> > ### Comment · Reviewer_JPZw · 2021-08-25
> > **Thanks for your clarification**
> >
> > Thanks for the detailed response! The experimental setting is clearer now, and I do not have more questions at this point.

---

### Official Review · Reviewer_SyKX · 2021-07-14

**Rating:** 6
**Confidence:** 4

**Summary:**

This paper proposes a new model for learning to execute programs on program-guided tasks. The model is based on transformer and uses cross-attention and self-attention mechanisms. It takes as inputs both the program structure (control flow information) and program semantics (through token embeddings), as well as task specifications. The model is evaluated on a visual reasoning benchmark and a 2D Minecraft RL benchmark. Experiments show that the proposed model achieves better performance than previous ad-hoc approaches for learning to execute programs in these domains.

**Limitations And Societal Impact:**

The limitations and societal impact is discussed in the Appendix.

**Main Review:**

This paper proposes a general approach for learning to execute programs on program-guided tasks. Using programs to guide learning is a promising direction and has received increasing attention, so the paper is targeting an important problem. The paper is well-written and easy to follow. The experimental evaluation is good, covering two distinct domains including visual reasoning and RL, and showed consistent improvement over prior ad-hoc approaches designed for each domain. Overall, this is a well-written paper.

Questions:

- What are the reasons for the performance improvement by ProTo over the prior ad-hoc approaches? Is it utilizing more information from the program, or just due to the better modeling power of a large transformer model? It would be helpful to see some analysis on the difference between ProTo and prior approaches to help understand why ProTo is better.

- There are some prior work that proposes general techniques for learning to execute programs, which is most closely related to the theme of the paper: Learning to Execute Programs with Instruction Pointer Attention Graph Neural Networks, David Bieber et.al. 2020.
It would be good if the authors can compare with these approaches.

- ProTo model feeds the last result matrix into the current step, therefore inducing a recurrent structure. How is this treated during training? Since for RL tasks with long horizon and only reward signal, back-propagating through the whole episode seems to be a high computation burden.



**Time Spent Reviewing:**

4

---

> ### Author Response · Authors · 2021-08-10
> **Reply to R2 SyKX: thank you!**
>
> We thank R2 for his positive and thoughtful review. We agree with your insightful positive opinions on the direction of program-guided learning. We also feel grateful for your appreciation of our novelty and clarity. Thanks again!
>
> Here we answer your questions:
>
> ### 1. Reasons for Improvements Over Ad-hoc Approaches
>
> We attribute the success of ProTo over prior ad-hoc approaches to the following reasons:
>
> 1. The strong representation power of the object-centric transformer architecture. By leveraging attention, ProTo can unify observations and the program guidance in an effective manner. The strong representation power of transformer has been validated and explored by other work in various domains (as discussed in Para 3 of Sec 2). We share the common belief with the recent transformer literature (e.g., Swin Transformer [1]) that the transformer could be a unified architecture used across tasks, which is one of the motivations of ProTo. In Table 1, we can observe transformer-based architecture outperforming CNN-based architectures by a considerable margin (>4%).
> 2. We have demonstrated our disentangled representation of the program enhances the power of transformers. In Table 2, we found that removing either structural guidance or semantic guidance would deteriorate ProTo's performance by >9% and >30% accordingly.
> 3. We propose the ProTo execution algorithm (Algorithm 1 and 2) to handle a complex DSL. In Table 4 and Figure 3, we show that a naive transformer cannot perform program-guided policy learning well. But ProTo can successfully outperform the strong baseline of PGA by leveraging the proposed execution algorithm with a disentangled representation of programs.
>
> ### 2. Comparison to IPA-GNN [2] (David Bieber et.al. 2020)
>
> Thanks for this valuable suggestion!  We carefully studied this recent paper as long as their released code. Although both ProTo and IPA-GNN are working on a similar topic, we want to highlight some major differences between ProTo and IPA-GNN:
>
> 1. **Motivation**: ProTo heads towards an important goal of neural symbolic AI that jointly develops the perception, planning, and decision ability, but IPA-GNN aims for better programming tools in software engineering.
> 2. **Task**: ProTo works on proposed program-guided tasks such as visual reasoning and policy learning, but IPA-GNN was designed for static analysis of codes.
> 3. **Model**: ProTo is based on transformer architecture with proposed disentangled program representation, while IPA-GNN is based on RNN with soft execution enabled by GNN.
>
> We ran their model based on their released code on the GQA dataset. The results are as follows. We also list the validation accuracy of approaches that are similar to IPA-GNN for the convenience of discussion. The training hyperparameters of mentioned approaches are the same (e.g., number of epochs, learning rates, and optimizers.). Some of the mentioned baselines consume the question as input.
>
> | Method       | Input          | Acc      |      |
> | ------------ | -------------- | -------- | ---- |
> | IPA-GNN [2]  | Program+Image  | 47.1     |      |
> | CNN-LSTM     | Question+Image | 46.6     |      |
> | BottomUp [3] | Question+Image | 49.7     |      |
> | Mac [4]      | Question+Image | 54.1     |      |
> | SNMN [5]     | Question+Image | 55.3     |      |
> | GAT Encoding | Program+Image  | 58.6     |      |
> | ProTo        | Program+Image  | **64.5** |      |
>
> We observe that IPA-GNN falls behind the SOTA on GQA by a large margin. This fact is not surprising for many reasons. First, the representation ability of RNN is not enough for cross-modality reasoning. Only very old baselines on the GQA benchmark leverages RNN-style modeling. Second, the GNN-style modeling of program structure cannot exploit the semantics and structural property of programs. Moreover, the architecture details of IPA-GNN are not optimized on GQA as well.
>
> We would add some of the above discussions to our revised paper. Thank you!
>
> ### 3. Recurrent Structure of Program Execution
>
> For your first question, one routine would finish execution if its ending conditions are met, as described in Sec 4.3. During training time, we follow the standard RL practice to penalize time-out exploration, as presented in Appendix D.2.1.
>
> As for the long-horizon and sparse reward problem, we agree that this is a tricky problem for the whole RL community. Currently, we do not observe a severe negative effect caused by the recurrent structure of the program execution. Probably it would be a problem when we scale up to a large map. Future work could incorporate the advances in large-scale game playing [6] to alleviate this problem.
>
> **References:**
>
> [1] Wenhu Chen, Zhe Gan, Linjie Li, Yu Cheng, William Wang, and Jingjing Liu. Meta module network for compositional visual reasoning. In *Proceedings of the IEEE/CVF Winter Conference on Applications of Computer Vision*, pages 655–664, 2021.
>
> [2] Bieber, David, et al. "Learning to execute programs with instruction pointer attention graph neural networks." *arXiv preprint arXiv:2010.12621* (2020).
>
> [3] Peter Anderson, Xiaodong He, Chris Buehler, Damien Teney, Mark Johnson, Stephen Gould, and Lei Zhang. Bottom-up and top-down attention for image captioning and visual question answering. In *Proceedings of the IEEE conference on computer vision and pattern recognition*, pages 6077–6086, 2018.
>
> [4] Drew A Hudson and Christopher D Manning. Compositional attention networks for machine reasoning. In *International Conference on Learning Representations*, 2018.
>
> [5] Ronghang Hu, Jacob Andreas, Trevor Darrell, and Kate Saenko. Explainable neural computation via stack neural module networks. In *Proceedings of the European conference on computer vision (ECCV)*, pages 53–69, 2018.
>
> [6] Vinyals, Oriol, et al. "Grandmaster level in StarCraft II using multi-agent reinforcement learning." *Nature* 575.7782 (2019): 350-354.

---

### Official Review · Reviewer_LuHc · 2021-07-15

**Rating:** 7
**Confidence:** 3

**Summary:**

This paper presents ProTo (Program-Guided Transformer), a modified
Transformer-based architecture that aims to learn a (fully neurally-parameterized)
 execution model for symbolic programs. The authors propose a parameterization scheme that encodes the program semantics (decomposed along minimal executable components of the symbolic program) and program syntax (using a transition mask defined based on the program tree); and describe multiple training regimes (varying from dense supervision based on ground truth or symbolic executions, to a fully RL-based distant supervision.)
The authors present empirical evaluations on two domains -- visual question answering (GQA) and policy execution (2D Minecraft) with respect to other comparable neural architectures that leverage symbolic program traces. They find comparable performance with respect to the SOTA, and demonstrate improvements in generalization to longer and novel programs.


**Ethical Concerns:**

I do not have significant ethical concerns about this paper.

**Limitations And Societal Impact:**

The authors provide a reasonable discussion of limitations and broader impact in the appendix (E and F).


**Main Review:**

*Strengths (soundness; theoretical grounding; empirical evaluation; significance and novelty; relevance to NeurIPS community)*:
- Application is interesting and of significant interest to the NeurIPS community. Domains used demonstrate promise for future work. Learning executors that can handle formal programming languages is an area of of interest with important potential applications for generalization and interpretability, and the authors demonstrate their model on two domains -- VQA and policy learning -- that highlight this. Further, while the authors focus on learning execution engines for formal program representations, it seems possible and of interest to test the generalizability of the same architecture (perhaps pre-trained on programs as a form of ‘synthetic data’ to natural language processing -- and indeed, the authors do approach this in their GQA approach (which uses a Seq2Seq model to perform the initial transformation from natural language into the DSL used here.)

The DSLs used are fairly complex -- the planning DSL has looping constructs.

- Architecture proposed is reasonably novel and lays the groundwork for future work. The proposed architecture is based on Transformers -- it uses attention blocks as its primary neural architectural component -- but takes interesting, reasonable seeming, (and I believe, novel) approaches to leverage the inherent semantic and syntactic structure of the program representation. This seems promising as an avenue for future work using similar architectures, and I think if the authors wished they could attempt to describe the most general instantiation of this architecture while signposting particular choices made in their concrete model that could be modified in the future (e.g the authors choose routines as minimal executable units, presumably for supervision purposes. But it seems like changes could be made so that `Filter(girl)` could be decomposed further for greater compositionality in the future.)

- Empirical evaluations show comparable performance to other SOTA architectures (like meta-module networks), and generalization experiments show significant performance improvements. The results from human-specified programs are particularly interesting. The authors compare against a good range of strong baselines and ablations.

For the generalization experiments, it would be nice to see a continuous graph (rather than bucketing the programs into standard testing, longer programs, and complex programs, for instance, in Table 4) that really shows the fall-off in performance between length of training programs vs. generalization performance -- perhaps this would even more clearly highlight the contributions of this model.

Similarly, the authors also present intriguing data using ‘unseen programs’ on the GQA dataset. I could personally use some clarification on how this worked -- was there additional ‘training’ at test time to learn to use the `verify` instances? Otherwise, how does the model know at all how to execute these primitives -- but it could be interesting and useful to see a similar continuous graph showing the fall-off in generalization performance as additional new primitives are introduced into the dataset, to really showcase compositionality or graceful robustness to new primitives.

*Weaknesses (soundness; theoretical grounding; empirical evaluation; significance and novelty; relevance to NeurIPS community)*:
- Discussion of generalization performance is promising but could be more rigorous, especially with respect to symbolic / formal baselines when they exist, or mixed neuro-symbolic methods. The authors mention several times that their approach -- learning the program executor entirely -- is based on prior work showing that learned executors often outperform symbolic ones.
This certainly may be the case on many domains of interest, and the authors convincingly show strong improvements over a model (NS-VQA) that uses a symbolic execution model on visual question answering, a domain that does seem to require some learnable component and that seems well-suited for this application.

On the second domain (Minecraft), however, we do in fact have a fully symbolic model -- the symbolic planner used to construct programs (and that provides the dense supervision.) In the strengths section above, I laid out some figures / generalization experiments that would help give a clearer picture of how this model performs relative to a fully-symbolic executor on two axes that seem most important to emulate: (a) modularity compositionality with respect to new program components introduced into the DSL; and (b) generalization to long programs and programs of arbitrary length.
Especially because the authors strongly suggest that a core goal of this model is to replace or outperform symbolic models on important future domains, I think it is important to have a clear picture of how well it can perform on fully symbolic tasks relative to symbolic baselines.

- Empirical performance in comparison to SOTA is comparable and competitive, but in many cases, except on generalization or complex programs, it is not very large. It suggests that a harder dataset would be necessary to really showcase the performance of this model, and on GQA, all models still seem to fall well short of human performance.
- (More minor): discussion of failure cases, especially wrt longer programs and control flow. Discussion of intermediate execution states and disentanglement. The GQA dataset does not have control flow. Loops and other more complex, characteristically programmatic structures are one area where I would have expected to see degraded generalization performance with respect to a symbolic model (e.g. performing more inaccurately with greater numbers of loops). The authors show results specifically on complex programs (with more than four If/While tokens) which are promising -- but again, I see this as an area that could have specifically probed where this model successfully learns to emulate the desired flow and when it may not. Perhaps some metric other than ground-truth accuracy -- such as an analysis of the intermediate execution state accuracy -- would also serve this purpose.
- Use of dense supervision. This is discussed in the limitations section (Appendix E), but raises questions about the scalability of this model with respect to its ultimate end goal. However, it could be interesting to see other ‘Partial Superivsion’ training regimes other than the one in the paper (knowing the final execution result of the whole program) -- what about supervision on some program components but not others (suggested by the GQA training regime -- which starts with a pre-trained object detector)? This could be an interesting area for testing compositionality and robustness to new program components as well.



**Time Spent Reviewing:**

2

---

> ### Author Response · Authors · 2021-08-10
> **Reply to R1 LuHc: thank you!**
>
> We thank the encouraging reviews from LuHc. We agree that ProTo can be scaled to be a general execution engine in the future for general programming languages. We totally agree that learning program executors across domains with complex DSLs is an important direction in this field. We feel delighted to agree that our work is a novel groundwork for future research in this field (actually, this is why we call our model "ProTo"). We feel happy to answer your questions as follows.
>
> ### 1. Continuous Graph of Generalization Experiments
>
> We thank your suggestions of drawing a continuous graph to visualize the generalization experiments. We did not draw this because of the spatial constraints of the paper. We will include this experiment in the Appendix of our paper in the next version.
>
> ### 2. Unseen Programs in the GQA dataset
>
> Similar to MMN[1], we design this generalization experiment to validate whether our model can be compositional to the program structure. Although the verify-type of questions are removed from the training split, the model still has a chance to see the "verify" token in the training split. For example, the program related to the logical question "Is the fruit on top of the tray yellow and thick?" requires two routines of ”verify(thick)“ and “verify(yellow)”. Note this question is not verify type and is not removed from the training splits. This example could validate whether the model compositionally learns to reason verify type questions. Please also note the general response for an explanation on this. We would also try to draw a continuous graph to illustrate the results of the generalization experiments.
>
> ### 3. Discussions about the Neural-Symbolic Baselines
>
> Thanks for your suggestions. The comparison between different neural symbolic approaches is a significant aspect of our work. On the first domain of visual reasoning, our paper shows that a mixed parametric neural-symbolic model would outperform pure symbolic non-parametric executors (Table 2). On the second domain of Minecraft, we have compared neural baselines (Figure 3 and Table 4). Indeed, we have a pure symbolic (non-learning) method: the planner. We would add more discussion and experiments to demonstrate the effectiveness of a learned executor beyond the symbolic planner:
>
> 1. The hard-coded planner requires a lot of ad-hoc engineering work to handle complex cases. But our method can learn from a sparse reward signal. For example, our method can learn to build a bridge to cross the river to fetch gold on the other side of the river without explicit supervision (just given a reward signal). However, a planner needs to handle this case with special treatment.
>
> 2. We experiment on MineCraft to validate the ability of ProTo to generalize across unseen routines. Specifically, we remove a routine from the training split (e.g., Mine(Gold)) and test on programs that contain the removed routine. This experiment validates the compositional generalization ability (e.g., generalize to Mine(Gold) after seeing Is_there(gold) and Mine(Silver)). Only a reward signal is used for training. Other experimental details are the same as described in Sec 5.2. The planner is set to choose a random legal action when meeting an unseen routine). The results on the validation dataset are presented as follows.
>
>    | Removed Routine | Planner Acc | ProTo Acc |      |
>    | --------------- | ----------- | --------- | ---- |
>    | Mine(Gold)      | 11.4        | 59.3      |      |
>    | Is_there(River) | 43.8        | 65.6      |      |
>    | Agent[Silver]   | 46.2        | 77.1      |      |
>
>    The results show that while our approach can reasonably generalize to unseen routines, the rule-based planner  does not get meaningful results when meeting unseen routines. We would add more systematical generalization experiments in the next version.
>
> 3. The planner cannot scale up to a large number of states. A planner cannot work well on a large-scale scenario such as the game GO [2].
>
> 4. The planner can never work on raw image observations. Note our transformer-based architecture can work on raw image input after dividing the observation into patches [3] or detecting objects in the raw image. But the symbolic planner has no way to work on raw image inputs.
>
> Thanks for this deep and thoughtful suggestion! We think these discussions and experiments would strengthen our claim on learning to execute programs. We do think the comparison between symbolic and neural methods is such a large and important topic that we cannot give a perfect answer in our paper. We hope a lot of future work can dive into this topic and give a better understanding.
>
> ### 4. Empirical Performance on Datasets
>
> We agree with you that both GQA and Minecraft have their issues (e.g., saturating performance of SOTAs), but they are the most suitable testbed for our approach at the moment. After the ProTo project, we will propose a comprehensive, novel, and large-scale dataset oriented on program-guided tasks. Thanks for your remark!
>
> ### 5. Discussions of Failures, Complex Programs, and More Metrics
>
> We have included visualized examples in Appendix Figure1-4. As for the failure cases of complex loops, we observe that ProTo fails when it needs to execute a lot of routines requiring a lot of steps (e.g., the agent needs to walk from one corner to another corner). We would visualize it via a video on the project page later. For the metrics, we stick to the same metrics with the program-guided agent (PGA) baseline in this paper. Your suggestion of proposing a fine-grained metric would be considered when we make our new dataset in future work.
>
> ### 6. The Use of Dense Supervision
>
> To help understand the performance of ProTo against baselines in the GQA and Minecraft, we follow the most common settings of neural symbolic approaches that leverage dense supervision or partial supervision. As for partial supervision, the most common setting on GQA assumes the training signal of the final reasoning step (the answer to the question). This setting is common because annotating the answer is much more feasible than annotating the full execution traces. It's interesting to experiment on other proportions of partial supervision, which would be added in the next version of our paper.
>
> Feel free to ask us anything about our paper during the discussion!
>
> **References:**
>
> [1] Wenhu Chen, Zhe Gan, Linjie Li, Yu Cheng, William Wang, and Jingjing Liu. Meta module network for compositional visual reasoning. In *Proceedings of the IEEE/CVF Winter Conference on Applications of Computer Vision*, pages 655–664, 2021.
>
> [2] Chen, Jim X. "The evolution of computing: AlphaGo." *Computing in Science & Engineering* 18.4 (2016): 4-7.
>
> [3] Dosovitskiy, Alexey, et al. "An image is worth 16x16 words: Transformers for image recognition at scale." *arXiv preprint arXiv:2010.11929* (2020).

---

> > ### Comment · Reviewer_LuHc · 2021-08-26
> > **Thank you!**
> >
> > Thank you for the very detailed response -- combined with the responses to the other reviewers, I have no further questions and will continue in the general discussion thread among reviewers above!

---

### Author Response · Authors · 2021-08-10
**General Response to Reviewers**

Thanks! We are excited that we received very positive, thoughtful, and insightful reviews from four expert reviewers. We are happy that all the reviewers appreciated the novelty of our approach and the significance of this direction. Here we summarize our contributions and potential impacts to the neural-symbolic field:

1. We formulate program-guided tasks, which require learning to execute the given program under task specifications. Program-guided tasks are heading towards a significant goal of neural-symbolic AI: building the communication between humans and machines via programs and learned executors.
2. We propose program-guided transformers (ProTo), which leverage the semantics and structures of the program to perform program-guided tasks. To the best of our knowledge, ProTo is the first model that leverages transformer-style architecture as learnable general executors in neural-symbolic tasks.
3. We conduct experiments on two exemplar program-guided tasks and demonstrate the state-of-the-art performance of ProTo.

We would highlight some improvements of our paper following reviewers' suggestions:

### Clarify the Unseen Programs Experiments

R1, R3, and R4 raise questions on the unseen programs experiments on GQA. In this experiment, we remove the verify-type programs. Note in GQA, the type of a program is determined by the last routine of the program. So even if we remove the verify-type programs, there are still verify-type routines in the training split, as long as these routines are not the last routines of the program. For example, for the following two data points on GQA:

1. **Question**: Does the napkin look red?

   **Program**: Select(Napkin) -> Verify(Red)

   **Program type**: verify-color

2. **Question**: Does the utensil on top of the table look clean and black?

   **Program**: Select(Table) -> RelateInv(Top)->Verify(Black)->Verify(Clean)->And()

   **Program type**: and

The first example is a verify-type, so it is removed in the training split. But the second example is not a verify-type, so it is contained in the training split. Note the model can see "verify" token during training, which makes it possible for generalization.

### Comparison to IPA-GNN

We compare and distinguish a new baseline IPA-GNN [1], which also studies learning to execute. See the response to R2 for details.

### Comparison to Planner

The comparison between neural, mixed neural-symbolic and symbolic approaches is a key point in our paper. Thanks to the suggestions from R1, we add a comparison to the symbolic planner in the Minecraft dataset. Refer to the response to R1 for details.

### Program Statistics and Human Program Details

We give more details of the GQA programs and the collected human-written programs. Results show that our collected human-written programs are more interesting, natural, and complex. See the response to R3 and R4 for details.

### Clarify the Method Part

We revise our papers to make the methods part clearer. We also provide more detailed explanations on the model input/output. We will also improve our model diagram. See the response to R4 for details.

### More Related Work and Fix Typos

We include more related work in the submission, and we fix some typos and grammar issues raised by R4. Thank you!

Thank you again and your ideas have made ProTo better! We are happy to discuss more with the insightful reviewers during the discussion period.

**References:**

 [1] Bieber, David, et al. "Learning to execute programs with instruction pointer attention graph neural networks." *arXiv preprint arXiv:2010.12621* (2020).

---

### Decision · Program_Chairs · 2021-09-27

**Decision:**

Accept (Poster)

**Comment:**

All reviewers appreciated the novel approach for learning to execute symbolic programs using both program syntax and semantics. Adding additional experimental evaluation and comparisons to IPA-GNN and symbolic planner for Minecraft was also greatly appreciated. It would be great to incorporate the feedback, and the experimental and architectural clarifications from the reviews and author responses in the final version.